# A New Algorithm for the Retrieval of Atmospheric Profiles from GNSS Radio Occultation Data in Moist Air and Comparison to 1DVar Retrievals

**Ying Li** [1,*]**, Gottfried Kirchengast** [2]**, Barbara Scherllin-Pirscher** [3]**, Marc Schwaerz** [2]**, Johannes K. Nielsen** [4]**, Shu-peng Ho** [5] **and Yun-bin Yuan** [1]

[1] State Key Laboratory of Geodesy and Earth's Dynamics, Institute of Geodesy and Geophysics (IGG), Chinese Academy of Sciences, 340 Xudong Road, Wuhan 430077, China; yybgps@asch.whigg.ac.cn
[2] Wegener Center for Climate and Global Change (WEGC) and Institute for Geophysics, Astrophysics, and Meteorology/Institute of Physics, University of Graz, Brandhofgasse 58010, Graz, Austria; gottfried.kirchengast@uni-graz.at (G.K.); marc.schwaerz@uni-graz.at (M.S.)
[3] Zentralanstalt für Meteorologie und Geodynamik (ZAMG), Hohe Warte, 381190 Vienna, Austria; barbara.scherllin-pirscher@zamg.ac.at
[4] Danish Meteorological Institute (DMI), Lyngbyvej 100, DK-2100 Copenhagen, Denmark; jkn@dmi.dk
[5] National Oceanic and Atmospheric Administration (NOAA), NESDIS/STAR/SMCD, Center for Weather and Climate Prediction, 5830 University Research Court, College Park, MD 20740-3818, USA; shu-peng.ho@noaa.gov
* Correspondence: liying@asch.whigg.ac.cn; Tel.: +86-186-2704-5396

**Abstract:** The Global Navigation Satellite System (GNSS) Radio Occultation (RO) is a key technique for obtaining thermodynamic profiles of temperature, humidity, pressure, and density in the Earth's troposphere. However, due to refraction effects of both the dry air and water vapor at low altitudes, retrieval of accurate profiles is challenging. Here we introduce a new moist air retrieval algorithm aiming to improve the quality of RO-retrieved profiles in moist air and including uncertainty estimation in a clear sequence of steps. The algorithm first uses RO dry temperature and pressure and background temperature/humidity and their uncertainties to retrieve humidity/temperature and their uncertainties. These temperature and humidity profiles are then combined with their corresponding background profiles by optimal estimation employing inverse-variance weighting. Finally, based on the optimally estimated temperature and humidity profiles, pressure and density profiles are computed using hydrostatic and equation-of-state formulas. The input observation and background uncertainties are dynamically estimated, accounting for spatial and temporal variations. We show results from applying the algorithm on test datasets, deriving insights from both individual profiles and statistical ensembles, and from comparison to independent 1D-Variational (1DVar) algorithm-derived moist air retrieval results from Radio Occultation Meteorology Satellite Application Facility Copenhagen (ROM-SAF) and University Corporation for Atmospheric Research (UCAR) Boulder RO processing centers. We find that the new scheme is comparable in its retrieval performance and features advantages in the integrated uncertainty estimation that includes both estimated random and systematic uncertainties and background bias correction. The new algorithm can therefore be used to obtain high-quality tropospheric climate data records including uncertainty estimation.

**Keywords:** GNSS atmospheric sounding; radio occultation; moist air retrieval; uncertainty propagation; algorithm validation

## 1. Introduction

The Global Navigation Satellite System (GNSS) Radio Occultation (RO) technique is an atmospheric sounding technique providing global-coverage, high-accuracy, and high-precision vertical profiles of Earth's atmosphere [1–5]. The technique uses receivers on Low Earth Orbit (LEO) satellites to receive GNSS signals after they propagated through the atmosphere in limb sounding geometry. Vertical profiling is achieved due to the satellites' orbital motions. As the signals propagate, they are bent due to the atmospheric refractivity gradients.

The accumulated bending angle can be calculated from precise orbit data and the excess phase measurements acquired by tracking the GNSS signals on a LEO satellite. The bending angle profile can in turn be converted to a refractivity profile using an Abel integration. In dry air conditions, atmospheric temperature, pressure, and density profiles can then be retrieved using a refractivity equation, hydrostatic integral, and ideal gas law [2].

In moist air conditions, however, which apply in the troposphere below about 9 km to 16 km, the refractivity is also significantly affected by moisture. In this case, there are four unknown variables, i.e., temperature, pressure, density, and humidity, but only three equations as stated above are available as constraint. This results in a temperature-humidity ambiguity problem [2,6–8] that fundamentally could only be solved by way of occultation technique extension by using higher frequency signals as proposed for microwave occultation [9–11]. Most air retrieval algorithms, which are the focus of this study, instead solve this problem for RO by way of data processing extension including background information on tropospheric temperature and/or humidity.

In early moist air retrieval algorithm designs, scientists used a direct method to retrieve tropospheric humidity or temperature profiles, by using background profiles of either temperature or humidity [2,6]. However, this method may induce sub-optimal uncertainty from background data assumed "exactly true". As a more general alternative, the one-dimensional variational (1DVar) method [12,13], also termed optimal estimation method [14], was suggested for moist-air retrieval [15] and further investigated by several studies [7,16–19].

The 1DVar method works by finding a maximum likelihood optimal estimate of a vertical atmospheric state profile $\mathbf{x}$, given a set of observations $\mathbf{y}_b$ and a priori knowledge on a background atmospheric state profile $\mathbf{x}_b$ as well as the error covariance matrices of both the observation and background information. The 1DVar can be written as a minimization of the following equation [20]:

$$J(\mathbf{x}) = \frac{1}{2}(\mathbf{x} - \mathbf{x}_b)^T \mathbf{B}^{-1}(\mathbf{x} - \mathbf{x}_b) + \frac{1}{2}(\mathbf{y}_o - \mathbf{H}[\mathbf{x}])^T \mathbf{O}^{-1}(\mathbf{y}_o - \mathbf{H}[\mathbf{x}]), \tag{1}$$

where $\mathbf{H}[\mathbf{x}]$ denotes a forward operator mapping the state $\mathbf{x}$ to the observation space $\mathbf{y}_o$. The matrices $\mathbf{B}$ and $\mathbf{O}$ are background and observation error covariance matrices, respectively, representing the standard uncertainties and correlations of the background data and the observation (plus forward-modeled) data. Minimizing the cost function $J(\mathbf{x})$ by variation of the state $\mathbf{x}$ yields the retrieved state $\mathbf{x}_r$ that minimizes the total deviation against background and observational data. The usual selection of $\mathbf{y}_o$ in moist-air retrieval by 1DVar is the observed refractivity profile from which temperature, humidity and surface pressure are retrieved as state $\mathbf{x}_r$ [17–20].

Currently, the RO data processing centers Constellation Observing System for Meteorology, Ionosphere, and Climate (COSMIC) Data Analysis and Archive Center (CDAAC), University Corporation for Atmospheric Research (UCAR) Boulder, Radio Occultation Meteorology Satellite Application Facility (ROM-SAF), Danish Meteorological Institute (DMI) Copenhagen, and National Oceanic and Atmospheric Administration (NOAA) Center for Satellite Applications and Research (STAR) Maryland, use 1DVar algorithm implementations for their (operational) moist air retrievals [20–25]. Both ROM-SAF and CDAAC moist air profiles are used for our evaluation of the new algorithm in this study.

ROM-SAF data used in this study are the latest reprocessed climate data records CDR v1.0, which are available at http://www.romsaf.org. The CDR v1.0 processing is based on ROPP 8.1 [26], with few adaptations. In its products, ROM-SAF Level 2B data provide moist-air profiles. These profiles are

retrieved by a 1DVar algorithm that uses retrieved refractivity and geometric altitude [27] together with background data from ERA-Interim (ERA-I) [28] as input. For each occultation event, the background temperature, specific humidity, and surface pressure are interpolated in time and space at 60 model levels from the ERA-I forecast available at a 3 h and 1° × 1° grid. The 1DVar configuration is defined by a few choices described in detail in the ROM-SAF 2018 report on Level 2B and 2C 1DVar products [20].

Refractivity uncertainty is parameterized as function of height. In the troposphere it is a straight line fixed at 0.2% at the tropopause and 2% at surface. Above the tropopause it is 0.2% but never below 0.02 N-units. Refractivity vertical correlations are assumed exponential with a 3 km correlation length. Background uncertainty is taken directly from the ERA-I error of first guess estimate provided by ECMWF. The surface pressure uncertainty has been inflated in order to adapt to an evident pressure difference in ERA-I forecast and analysis. The ERA-I vertical uncertainty profiles are averaged into 5° latitude bins while the vertical correlations are provided by ECMWF as a fixed correlation matrix.

CDAAC-provided moist air profiles used in this study are the reprocessed wetPrf data records of the Challenging Minisatellite Payload CHAMP and COSMIC RO missions (CHAMP data: 2016.2430 version, available online via CDAAC website; COSMIC: update of 2013.3520 version, available online via CDAAC in future). At CDAAC, background profiles are taken from ECMWF gridded low-resolution analysis data collocated to RO locations [29]. The observation uncertainties include both systematic and random components, which are latitude-dependent and are estimated from the statistics of innovation vectors for a reasonably long period such as one month, and the information was updated regularly from the statistics for a recent period to yield best-possible performance [30]. The estimation of the background uncertainties at CDAAC is similar to the estimation of observation uncertainties. The correlation matrix was estimated using a fifth-order correlation function, such as used by Steiner and Kirchengast [31], which is similar to a Gaussian function in shape but compactly supported (in a mathematical sense).

The formulation of **B** and **O** is critical for the moist air retrieval, since it determines the weights of background and observed data that lead to the formally optimal profiles according to Equation (1). The 1DVar method is successful and retrieved moist profiles have been used in several climate and weather studies and good results were obtained [32–34].

In this study, we introduce a "linearized 1DVar" moist air retrieval algorithm as a robust 1DVar alternative that sequentially combines the direct method with optimal estimation. The new method is designed to derive tropospheric temperature, humidity, and pressure profiles at the same quality as 1DVar, and provides a robust linear non-iterative propagation chain, including "direct method" humidity and temperature profile retrievals as interim results, and transparent and comprehensive uncertainty estimates. It includes empirical models of background and observation uncertainties, to optimally determine the weights of background and observed data. The new algorithm was initially motivated, designed, and theoretically derived by Kirchengast et al. [35]. It is introduced here in detail in its current updated form, together with the formulation of its input ingredients, including the uncertainty formulations involved.

The new scheme is implemented since 2013 already—in line with its initial design with refractivity-equation closure for pressure retrieval [35] and in a basic form with static input uncertainty profiles—in the Wegener Center for Climate and Global Change (WEGC) current Occultation Processing System version 5.6 (OPS v5.6). It has shown reliable results for entire climate records in several studies [24,25,36–38].

In this study we denote this initial implementation using static uncertainties the "OPSv5.6 approach", while we denote the updated form that uses dynamic input uncertainties, an equation-of-state closure for pressure retrieval, and forecast-minus-analysis bias correction of background profiles the "dynamic approach". The advanced inclusions of propagating full covariance matrices as well as estimated systematic uncertainty and observation-to-background weighting ratio profiles, as implemented in the new rOPS system [39–44], are beyond the scope of this study and will be introduced in a separate follow-on paper.

Both the OPSv5.6 and the dynamic approach are tested using exemplary ensembles of simulated Meteorological Operational (MetOp) satellite data as well as real-observed CHAMP and COSMIC data

and evaluated in a retrieval performance validation with corresponding profile ensembles from 1DVar moist air retrievals provided by ROM-SAF and CDAAC.

The paper is structured as follows: Section 2 describes the new algorithm in terms of the algorithm basis, detailed algorithm steps for profile retrieval and uncertainty estimation, and explains the retrieval scheme and process. Section 3 presents the algorithm evaluation results in terms of its performance for individual RO events as well as of its statistical performance in different latitude bands. Finally, a summary and conclusions are given in Section 4. Appendices A–C provide complementary information on aspects of numerical implementation, background bias correction, and vertical error correlations.

## 2. Methodology—The New Moist Air Retrieval Algorithm

### 2.1. Algorithm Basis

In dry air condition, the refractivity $N$ can be expressed as $N = c_1 R \rho_d = c_1(p_d/T_d)$, where $R = 287.06$ J kg$^{-1}$ K$^{-1}$ is the dry air gas constant [2,35], $c_1 = 77.60$ K hPa$^{-1}$ is the Smith-Weintraub refractivity formula first constant ("dry term"), and $\rho_d$, $p_d$, and $T_d$ are RO-retrieved dry density, dry pressure, and dry temperature, respectively. The profiles $p_d(z)$ and $T_d(z)$ are derived in RO processing by the so-called dry air retrieval step, using the hydrostatic integral and the equation of state (e.g., [2,41]), and are available as input to the moist air retrieval.

The refractivity formula embodying the dry air equation of state $p_d/\rho_d = RT_d$, allows formulating the ratio of $p_d$ and $T_d$ in terms of generic refractivity at any altitude level of $z$:

$$c_1 \frac{p_d(z)}{T_d(z)} = N(T(z), V_w(z), p(z)), \tag{2}$$

wherein $T$ and $p$ represent physical (moist) temperature and pressure, respectively, and $V_w$ is the water vapor volume mixing ratio. The latter relates to pressure $p$, water vapor partial pressure $e$, and specific humidity $q$ as:

$$V_w(z) = \frac{e(z)}{p(z)} = \frac{q(z)}{a_w + b_w q(z)}, \tag{3}$$

where $a_w = 0.622$ is the moist air gas constant ratio, $b_w = 1 - a_w = 0.378$ is the moist air gas constant ratio complement [35,45].

The generic refractivity on the right hand side (R.H.S.) of Equation (2) denotes any existing type of refractivity relation from Smith-Weintraub type to Thayer type, with any given coefficients [46–50], and can be expressed in the form:

$$N(T(z), V_w(z), p(z)) = N(z) = c_1 \frac{p(z)}{T(z)} (f_0 + f_1 \cdot V_w(z)), \tag{4}$$

where $f_0$ is unity or close to unity and $f_1$ is close to $(c_2/T)/c_1$, where $c_2 = 3.73 \times 10^5$ K$^2$ hPa$^{-1}$ represents the Smith-Weintraub refractivity formula second constant ("wet term"). The exact values of $f_0$ and $f_1$ depend on which refractivity formula is used and whether ideal gas behavior is adopted. As the current OPSv5.6 and rOPS baseline, the standard Smith-Weintraub refractivity formula is used, corresponding to $f_0 = 1$ and $f_1 = (c_2/T)/c_1 = c_T/T$ [K] and $c_T = c_2/c_1 = 4806.7$ K. These values are used later on.

Healy [48] conveys that this standard relation continues to be a very good representation and its use keeps parametric consistency with other processing chains using it as well. Kirchengast et al. [35] explain that the perturbations to $f_0$ and $f_1$ will be very small (order $10^{-3}$ or smaller) for any more advanced refractivity formulation so that they could be readily added as "epsilon terms" within the step 1a / step 1b iteration algorithm (cf. Section 2.2) if desired. Aparicio and Laroche [50] caution that any use of an advanced refractivity formulation beyond the Smith-Weintraub form should also consistently use a correspondingly advanced equation-of-state formulation accounting for non-ideal gas behavior; an aspect that can as well be accounted for by adding "epsilon terms" in the current algorithm.

Using the R.H.S. of Equation (4) equated with the left-hand-side (L.H.S.) of Equation (2) as basis to explicitly express moist air profiles of $T$ and $V_w$, we get the following two mutually equivalent forms:

$$T(z) = T_d(z)\frac{p(z)}{p_d(z)}(f_0 + f_1 \cdot V_w(z)), \tag{5}$$

$$V_w(z) = \frac{\frac{p_d(z)}{p(z)}T(z) - f_0 \cdot T_d(z)}{f_1 \cdot T_d(z)}. \tag{6}$$

Based on hydrostatic integration, the dry pressure and the moist pressure at any given altitude level $z$ can be expressed as:

$$\frac{d \ln p_d(z)}{dz} = -\frac{g(z)}{RT_d(z)}, \text{ and} \tag{7}$$

$$\frac{d \ln p(z)}{dz} = -\frac{g(z)}{RT(z)(1 + c_w q(z))}, \tag{8}$$

where $c_w = 1/a_w - 1 = 0.608$ is the moist air humidity coefficient for virtual temperature [35]. By expressing the moist pressure vertical increment $dlnp$ in terms of the dry pressure increment $dlnp_d$, and also using Equation (3) to convert $q$ to $V_w$, we get:

$$d \ln p(z) = \beta(z) \cdot d \ln p_d(z), \tag{9}$$

$$\text{where } \beta(z) = \frac{T_d(z)(1 + b_w V_w(z))}{T(z)(1 + 2b_w V_w(z))}. \tag{10}$$

Since the differential increments $dlnp$ and $dlnp_d$ will be log-linearly discretized over adjacent levels, we can write $dlnp = dlnp(z_i) - dlnp(z_{i-1}) = \ln[p(z_i)/p(z_{i-1})]$, where $i$ represents the corresponding level indices. Similarly, we can write $dlnp_d = \ln[p_d(z_i)/p_d(z_{i-1})]$ for the dry pressure increment. Based on these expressions, we can then derive the expression of $p$ at any altitude level $z_i$ as:

$$p(z_i) = p(z_{i-1})\left(\frac{p_d(z_i)}{p_d(z_{i-1})}\right)^{\beta(z_{i-1/2})}, \tag{11}$$

where $\beta(z_{i-1/2}) = \frac{T_d(z_i) + T_d(z_{i-1})}{T(z_i) + T(z_{i-1})} \cdot \frac{1 + b_w \sqrt{V_w(z_i)V_w(z_{i-1})}}{1 + 2b_w \sqrt{V_w(z_i)V_w(z_{i-1})}}$ represents the exponent of fractional dry pressure change between levels $z_i$ and $z_{i-1}$ that leads to matching this change to the fractional moist pressure change. Since $T_d$ is always smaller than $T$ if moisture is non-zero, $\beta$ is (slightly) smaller than one, expressing that $p$ is changing less than $p_d$, consistent with the fact that $p_d$ is always larger than $p$ for non-zero moisture [51]. The specific formulation of $\beta$, with temperature expressed as mid-layer linear average (arithmetic mean) between the two levels, and water vapor mixing ratio as mid-layer log-linear average (geometric mean), is found helpful for high numerical accuracy at any given level spacing.

Based on these general expressions of Equations (5), (6), and (11), we can either solve for $T$ and $p$ if $q$ is prescribed, or for $V_w$ (and hence $q$ via Equation (3)) and $p$ if $T$ is prescribed. We can do this by a simple iteration at any arbitrary altitude level $z$ where a suitably adjacent level has been solved for $p$ before (starting at a "tropospheric top" level with negligible moisture where $p_d$ essentially equals $p$). If $q$ (and hence $V_w$) is prescribed, then, for any altitude level, we iterate the pairs of Equations (5) and (11) until $T$ has converged to within a small tolerance $dT_{tol}$, and $p$ will be consistent with the converged $T$.

Similarly, if $T$ is prescribed, we iterate the pair of Equations (6) and (11) until $V_w$ has converged to within a small tolerance $(dV_w/V_w)_{tol}$, and $p$ will then be consistent with the converged $V_w$. This formulation of the "direct method" of moist air retrieval is highly robust and versatile and applicable to arbitrary non-equidistant vertical grids of any level number (from minimum two levels) and vertical range from a chosen "tropospheric top" level to bottom of profile.

Given the resulting temperature and humidity profiles as well as estimates of their uncertainties and of the background profile uncertainties, we may then proceed to an optimally estimated profile for each, temperature and humidity, by combining these profiles with their corresponding background profiles in an inverse-variance-weighted manner. The following section provides more details.

*2.2. Algorithm Description*

The scheme and sequence of the new moist air retrieval algorithm is shown in Figure 1. The method consists of three steps. The first step includes two (formally parallel) sub-steps, which are independent from each other.

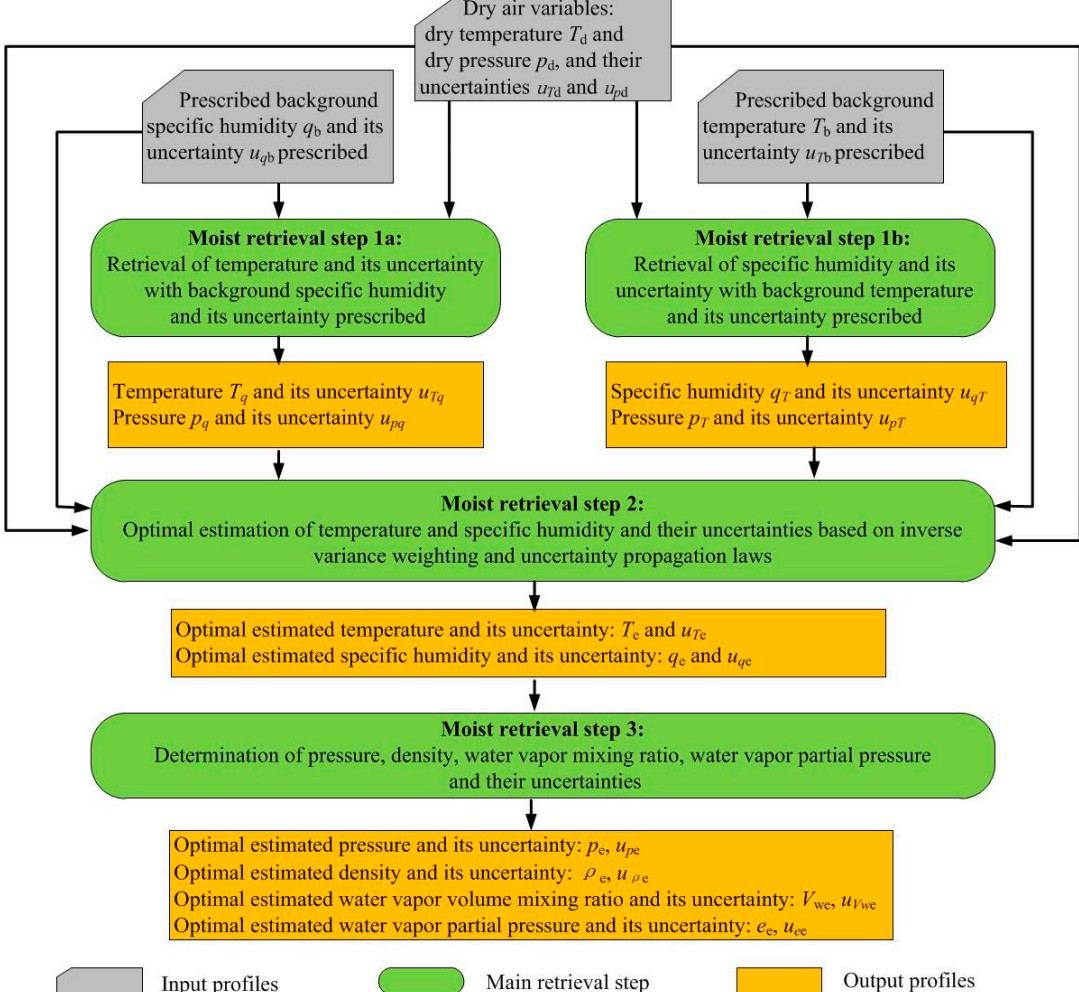

**Figure 1.** Schematic illustration of the algorithmic steps of the new moist air retrieval algorithm, for description see Section 2.

Step 1a is to retrieve temperature and pressure as well as their associated uncertainties with specific humidity and its uncertainty prescribed. Step 1b is to retrieve specific humidity and pressure as well as their associated uncertainties with temperature and its uncertainty prescribed.

Step 2 is to combine the retrieved temperature profile from step 1a and humidity profile from step 1b with their corresponding background profiles based on inverse variance weighting in order to obtain optimally estimated temperature and humidity profiles. This core step serves to eliminate the effects from sub-optimal estimation using fixed prescribed profiles by optimally weighting retrieved and background profiles each for temperature and humidity so as to arrive at a best estimate consistent with the available input uncertainty knowledge.

Step 3 then calculates optimal pressure, density, water vapor volume mixing ratio, and water vapor partial pressure profiles based on the optimally estimated temperature and specific humidity profiles from step 2. It uses standard thermodynamic relations for the purpose, such as hydrostatic integration according to Equation (8), and the most air equation of state.

The algorithm, as used in the OPSv5.6 implementation and in this study, focuses on the altitude range from a "tropospheric top" level of 16 km downward and hence covers the entire range where specific humidity may be non-negligibly small [51]. ECMWF operational 24h forecast fields are used to provide prescribed background temperature and specific humidity profiles collocated to the analyzed RO events. Below we introduce these three steps in detail, with some specific but relevant aspects of numerical implementation of steps 1a and 1b described in Appendix A.

*Step 1a: Retrieval of temperature and its uncertainty with specific humidity prescribed*

The input profiles of this step include: the prescribed background specific humidity $q_b(z)$ and its uncertainty $u_{qb}(z)$; the observed dry temperature $T_d(z)$ and its uncertainty $u_{Td}(z)$; and the observed dry pressure $p_d(z)$ and its uncertainty $u_{pd}(z)$. The output profiles include: temperature $T_q(z)$ and its uncertainty $u_{Tq}(z)$; pressure $p_q(z)$ and its uncertainty $u_{pq}(z)$, where the subscript $q$ denotes variables retrieved with specific humidity prescribed. Using the prescribed $q_b(z)$, the corresponding water vapor volume mixing ratio $V_{wb}(z)$ can be calculated using Equation (3). Then, based on Equations (5) and (11), $T_q(z)$ and $p_q(z)$ at altitude level $z_i$ can be expressed as:

$$T_q(z_i) = T_d(z_i)\frac{p_q(z_i)}{p_d(z_i)}\left(1 + \frac{c_T}{T_q(z_i)}\cdot V_{wb}(z_i)\right), \text{ and} \tag{12}$$

$$p_q(z_i) = p_q(z_{i-1})\left(\frac{p_d(z_i)}{p_d(z_{i-1})}\right)^{\beta_q(z_{i-1/2})}, \tag{13}$$

where $\beta_q(z_{i-1/2}) = \frac{T_d(z_i)+T_d(z_{i-1})}{T_q(z_i)+T_q(z_{i-1})}\cdot\frac{1+b_w\sqrt{V_{wb}(z_i)V_{wb}(z_{i-1})}}{1+2b_w\sqrt{V_{wb}(z_i)V_{wb}(z_{i-1})}}$. For each altitude $z_i$ from the initial altitude of our moist retrieval $z_{iniMoist}$ = 16 km to the bottom level of the profile, iteration over Equations (12) and (13) yields the profiles of $T_q$ and $p_q$.

The variance of $T_q(z)$ denoted as $u_{Tq}^2(z)$ can be obtained from propagating the variance profiles $u_{Td}^2(z)$ and $u_{qb}^2(z)$ based on a linearized version of Equation (12), linearized with some reasonable assumptions (cf. Appendix A):

$$u_{Tq}^2(z) = \left(\frac{p_q(z)}{p_d(z)}\right)^2 u_{Td}^2(z) + \left(\frac{p_q(z)}{p_d(z)}\frac{T_d(z)}{T_q(z)}c_{q2T}\right)^2 u_{qb}^2(z), \tag{14}$$

where $c_{q2T}$ = 7727.9 K is the moist air humidity coefficient in temperature error estimation and the square-root of $u_{Tq}^2(z)$ is the desired uncertainty profile.

Similarly, based on a linearized version of Equation (13), uncertainty of $p_q(z)$ denoted as $u_{pq}(z)$ can be calculated as:

$$u_{pq}(z) = \beta_q(z)\left(\frac{p_q(z)}{p_d(z)}\right)u_{pd}(z), \tag{15}$$

where $\beta_q(z) = \frac{T_d(z)(1+b_w V_{wb}(z))}{T_q(z)(1+2b_w V_{wb}(z))}$.

*Step 1b: Retrieval of specific humidity and its uncertainty with background temperature prescribed*

The input profiles of this step include: prescribed temperature $T_b(z)$ and its uncertainty $u_{Tb}(z)$; observed dry temperature $T_d(z)$ and its uncertainty $u_{Td}(z)$; and observed dry pressure $p_d(z)$ and its uncertainty $u_{pd}(z)$. The output profiles include the specific humidity $q_T(z)$ and its uncertainty $u_{qT}(z)$ and pressure $p_T(z)$ and its uncertainty $u_{pT}(z)$. According to Equations (6) and (11), the corresponding water vapor volume mixing ratio profile $V_{wT}(z)$ and $p_T(z)$ can be expressed as:

$$V_{wT}(z_i) = \frac{\frac{p_d(z_i)}{p_T(z_i)}T_b(z_i) - T_d(z_i)}{c_T \cdot \frac{T_d(z_i)}{T_b(z_i)}}, \text{ and} \tag{16}$$

$$p_T(z_i) = p_T(z_{i-1})\left(\frac{p_d(z_i)}{p_d(z_{i-1})}\right)^{\beta_T(z_{i-1/2})}, \tag{17}$$

where $\beta_T(z_{i-1/2}) = \frac{T_d(z_i)+T_d(z_{i-1})}{T_b(z_i)+T_b(z_{i-1})} \cdot \frac{1+b_w\sqrt{V_{wT}(z_i)V_{wT}(z_{i-1})}}{1+2b_w\sqrt{V_{wT}(z_i)V_{wT}(z_{i-1})}}$, where $c_T = c_2/c_1 = 4806.7$ K has been described above. $V_{wT}(z)$ and $p_T(z)$ can be solved by iterating level by level top-downward from $z_i = z_{\text{iniMoist}}$ to the bottom level. After obtaining $V_{wT}(z)$, the corresponding specific humidity $q_T(z)$ can be calculated using an inverse version of Equation (3) and its variance $u_{qT}^2$ can be propagated from the variance profiles of $u_{Tb}^2(z)$ and $u_{Td}^2(z)$ based on a linearized version of Equation (16) (cf. Appendix A):

$$u_{qT}^2(z) = \left(\frac{2\frac{p_d(z)}{p_T(z)}T_b(z) - T_d(z)}{T_d(z)}c_{T2q}\right)^2 u_{Tb}^2(z) + \left(\frac{\frac{p_d(z)}{p_T(z)}T_b^2(z)}{T_d^2(z)}c_{T2q}\right)^2 u_{Td}^2(z). \tag{18}$$

The square root of $u_{qT}^2(z)$ is the desired uncertainty profile $u_{qT}(z)$.

Similarly, based on a linearized version of Equation (17), the uncertainty of pressure $u_{pT}(z)$ can be calculated as:

$$u_{pT}(z) = \beta_T(z)\left(\frac{p_T(z)}{p_d(z)}\right)u_{pd}(z), \tag{19}$$

where $\beta_T(z) = \frac{T_d(z)(1+b_w V_{wT}(z))}{T_b(z)(1+2b_w V_{wT}(z))}$.

*Step 2: Optimal estimation of temperature and specific humidity and their uncertainties*

Based on the retrieved temperature profile $T_q(z)$ and its uncertainty $u_{Tq}(z)$ obtained in step 1a and also on the prescribed background temperature profile $T_b(z)$ and its uncertainty $u_{Tb}(z)$, the optimally estimated temperature profile $T_e(z)$ can be calculated by combining $T_q(z)$ and $T_b(z)$ based on inverse variance weighting at all altitude levels:

$$T_e(z) = \left(\frac{u_{Tb}^2(z)}{u_{Tq}^2(z) + u_{Tb}^2(z)}\right)T_q(z) + \left(\frac{u_{Tq}^2(z)}{u_{Tq}^2(z) + u_{Tb}^2(z)}\right)T_b(z). \tag{20}$$

Furthermore, its variance profile $u_{Te}^2(z)$ can be estimated using the uncertainty propagation law as:

$$u_{Te}^2(z) = \left(\frac{1}{u_{Tq}^2(z)} + \frac{1}{u_{Tb}^2(z)}\right)^{-1} = \frac{u_{Tq}^2(z)u_{Tb}^2(z)}{u_{Tq}^2(z) + u_{Tb}^2(z)}, \tag{21}$$

where the square-root of $u_{Te}^2(z)$ is the desired uncertainty profile $u_{Te}(z)$.

Similarly, using the retrieved specific humidity profile $q_T(z)$ and its uncertainty $u_{qT}(z)$ obtained in step 1b, and also on the prescribed specific humidity profile $q_b(z)$ and its uncertainty $u_{qb}(z)$, the optimally estimated specific humidity profile $q_e(z)$ and its variance $u_{qe}^2(z)$ can be estimated as:

$$q_e(z) = \left(\frac{u_{qb}^2(z)}{u_{qT}^2(z) + u_{qb}^2(z)}\right)q_T(z) + \left(\frac{u_{qT}^2(z)}{u_{qT}^2(z) + u_{qb}^2(z)}\right)q_b(z), \text{ and} \tag{22}$$

$$u_{qe}^2(z) = \left(\frac{1}{u_{qT}^2(z)} + \frac{1}{u_{qb}^2(z)}\right)^{-1} = \frac{u_{qT}^2(z)u_{qb}^2(z)}{u_{qT}^2(z) + u_{qb}^2(z)}, \tag{23}$$

where again the square-root of $u_{q\mathrm{e}}^2$ is the desired uncertainty profile $u_{q\mathrm{e}}(z)$.

*Step 3: Optimally estimated pressure, density, water vapor volume mixing ratio, water vapor partial pressure, and their associated uncertainties*

Based on the optimally estimated temperature and specific humidity profiles, the optimally estimated water vapor volume mixing ratio $V_{\mathrm{we}}(z)$, pressure $p_{\mathrm{e}}(z)$, density $\rho_{\mathrm{e}}(z)$, and water vapor partial pressure $e_{\mathrm{e}}(z)$ can be calculated quite straightforwardly since the relevant retrieval operators are known. The corresponding uncertainty profiles $u_{V\mathrm{we}}(z)$, $u_{p\mathrm{e}}(z)$, $u_{\rho\mathrm{e}}(z)$, and $u_{e\mathrm{e}}(z)$, can also be calculated using variance-based uncertainty propagation, given the state retrieval operators.

Using the optimally estimated specific humidity profile $q_{\mathrm{e}}(z)$, the derived water vapor volume mixing ratio $V_{\mathrm{we}}(z)$ can be calculated according to Equation (3):

$$V_{\mathrm{we}}(z) = \frac{q_{\mathrm{e}}(z)}{a_{\mathrm{w}} + b_{\mathrm{w}} q_{\mathrm{e}}(z)}, \tag{24}$$

and its associated uncertainty can be calculated, based on linearization of Equation (24), according to the uncertainty propagation law:

$$u_{V\mathrm{we}}(z) = \frac{a_{\mathrm{w}}}{\left(a_{\mathrm{w}} + b_{\mathrm{w}} q_{\mathrm{e}}(z)\right)^2} u_{q\mathrm{e}}(z). \tag{25}$$

The optimally estimated pressure profile $p_{\mathrm{e}}(z)$ can be calculated using the temperature profile $T_{\mathrm{e}}(z)$ and volume mixing ratio profile $V_{\mathrm{we}}(z)$ based on Equation (11):

$$p_{\mathrm{e}}(z_i) = p_{\mathrm{e}}(z_{i-1})\left(\frac{p_{\mathrm{d}}(z_i)}{p_{\mathrm{d}}(z_{i-1})}\right)^{\beta_{\mathrm{e}}(z_{i-1/2})}, \tag{26}$$

with $\beta_{\mathrm{e}}(z_{i-1/2}) = \frac{T_{\mathrm{d}}(z_i)+T_{\mathrm{d}}(z_{i-1})}{T_{\mathrm{e}}(z_i)+T_{\mathrm{e}}(z_{i-1})} \cdot \frac{1+b_{\mathrm{w}}\sqrt{V_{\mathrm{we}}(z_i)V_{\mathrm{we}}(z_{i-1})}}{1+2b_{\mathrm{w}}\sqrt{V_{\mathrm{we}}(z_i)V_{\mathrm{we}}(z_{i-1})}}$. This calculation, started at the $z_{\mathrm{iniMoist}}$ level as the previous pressure retrievals, is effectively based on the hydrostatic equation (Equation (7) or (8)) (in the convenient variant available in the context of this algorithm) and provides a pressure profile hydrostatically consistent with the estimated temperature and humidity profiles. We call this a hydrostatic-equation-based closure scheme for the retrieval of pressure to emphasize that it is improved over the refractivity-equation-based closure scheme used in the OPSv5.6 approach.

In the OPSv5.6 approach being part of the OPSv5.6 processing system, the pressure profile is derived as $p_{\mathrm{e}}(z) = p_{\mathrm{d}}(z)\frac{T_{\mathrm{e}}(z)}{T_{\mathrm{d}}(z)(1+c_2/(c_1 T(z))\cdot V_{\mathrm{we}}(z))}$, which is based on the Smith-Weintraub equation and implies that pressure is consistent with refractivity, temperature, and humidity. Due to errors in the refractivity profile, this pressure profile is somewhat "noisy" against the hydrostatic pressure profile that is fully consistent with the retrieved temperature and humidity.

The uncertainty of the estimated pressure profile can be calculated using a linearized version of Equation (26) in the form:

$$u_{p\mathrm{e}}(z) = \beta_{\mathrm{e}}(z)\frac{p_{\mathrm{e}}(z)}{p_{\mathrm{d}}(z)} u_{p\mathrm{d}}(z), \tag{27}$$

where $\beta_{\mathrm{e}}(z) = \frac{T_{\mathrm{d}}(z)(1+b_{\mathrm{w}} V_{\mathrm{we}}(z))}{T_{\mathrm{e}}(z)(1+2b_{\mathrm{w}} V_{\mathrm{we}}(z))}$.

Using Equation (3), the water vapor partial pressure profile $e_{\mathrm{e}}(z)$ can be computed as:

$$e_{\mathrm{e}}(z) = V_{\mathrm{we}}(z)p_{\mathrm{e}}(z), \tag{28}$$

and its variance profile can be calculated as:

$$u_{\mathrm{e}e}^2(z) = p_{\mathrm{e}}^2(z)u_{V\mathrm{we}}^2(z) + V_{we}^2 u_{p\mathrm{e}}^2(z). \tag{29}$$

The square root of this variance profile is the uncertainty profile $u_{ee}(z)$.

Finally, the density profile $\rho_e(z)$ can be derived by using the equation of state in moist air:

$$\rho_e(z) = \frac{p_e(z)}{RT_e(z)(1 + c_w q_e(z))},$$

(30)

and, based on a linearized version of Equation (30), its variance profile can be calculated as:

$$u_{\rho e}^2(z) = \left(\frac{1}{RT_e(z)(1+c_w q_e(z))}\right)^2 u_{pe}^2(z) + \left(\frac{p_e(z)}{RT_e^2(z)(1+c_w q_e(z))}\right)^2 u_{Te}^2(z)$$
$$+ \left(\frac{c_w p_e}{RT_e(z)(1+c_w q_e(z))^2}\right)^2 u_{qe}^2(z)$$

(31)

Again the square root of the variance profile is the desired uncertainty profile $u_{\rho e}(z)$.

### 2.3. Modeling of Observation and Background Uncertainties

The uncertainties of observed and background / prescribed variables are key for determining their weights in the optimally estimated profiles and are therefore critical for providing accurate moist profiles and associated uncertainty estimates. In the new algorithm, we dynamically estimate the background and observation uncertainties. As evident from above, these uncertainties include the background temperature uncertainty profile $u_{Tb}(z)$, the background specific humidity uncertainty profile $u_{qb}(z)$, the observed dry temperature uncertainty profile $u_{Td}(z)$, and the observed dry pressure uncertainty profile $u_{pd}(z)$. We sequentially describe below how we estimated these uncertainties for this study.

### 2.3.1. Observation Uncertainty Modeling

The observation uncertainty of both observed dry temperature $u_{Td}$ and observed dry pressure $u_{pd}$ are modeled following the empirically derived error model developed by Scherllin-Pirscher et al. [51]. Currently, both the OPSv5.6 and the dynamic approach use this model to estimate the observation uncertainty. In the future rOPS system, the propagated individual-profile based observation uncertainties (and error correlation matrices) will be used [43].

The model structure is the same for both temperature and humidity, only with different parameter settings. We set the parameters based on our tests for moist air retrieval, close to original ones of Scherllin-Pirscher et al. [51]. The vertical structure of the model, needed here only up to the bottom of the stratosphere, is:

$$s_{\text{model}}(z) = \begin{cases} s_0 + q_0\left[\frac{1}{z^p} - \frac{1}{z_{\text{Ttop}}^p}\right] & \text{for } z \leq z_{\text{Ttop}} \\ s_0 & \text{for } z_{\text{Ttop}} \leq z < z_{\text{Sbot}} \end{cases},$$

(32)

where $z_{\text{Ttop}}$ is the top altitude of the troposphere domain, $z_{\text{Sbot}}$ is the bottom altitude of the stratosphere domain, $s_0$ is the standard error (uncertainty) in the upper troposphere/lower stratosphere domain, $q_0$ is the best-fit magnitude parameter for the tropospheric model, and $p$ is the associated exponent parameter. The complete parameter settings are summarized in Table 1.

**Table 1.** Parameter settings for the observational uncertainty model for dry temperature and dry pressure.

|          | $z_{\text{Ttop}}$ | $z_{\text{Sbot}}$ | $s_0$  | $q_0$          | $p$ |
|----------|-------------------|-------------------|--------|----------------|-----|
| $u_{Td}$ | 10.0 km           | 20.0 km           | 0.7 K  | 3 K km$^p$     | 0.5 |
| $u_{pd}$ | 10.0 km           | 17.0 km           | 0.15%  | 0.7 %km$^p$    | 0.5 |

### 2.3.2. Background Uncertainty Modeling

The calculation follows the approach of Li et al. [52,53] and starts from the preparation of a daily updated global three-dimensional (latitude, longitude, and altitude) background uncertainty fields. The horizontal grid resolution of the uncertainty fields is 10° latitude × 20° longitude (center of base cell is 5°N, 10°E). The vertical resolution was updated to 100 m level spacing from 0.1 km to 20 km altitude. This construction yields the daily uncertainty fields at a global 18 × 18 × 200 grid. All the mean basic profiles that are required for the calculation of uncertainties are saved for each 10° × 20° grid cell center location and on the defined 200 vertical grid levels.

In each 10° × 20° grid cell and on the 200-level standard vertical grid, several types of basic variables are pre-calculated and saved for both background temperature and specific humidity. These basic variables include (the same notations of some basic variables are used for both temperature and humidity due to the same calculation method): (1) the mean analysis profile of temperature $\overline{T}_a$ and specific humidity $\overline{q}_a$; (2) the standard deviations of the ensemble of analysis values against its mean $s_a$; (3) bias of the mean analysis profile $b_a$; (4) the mean forecast profile of temperature $\overline{T}_f$ and specific humidity $\overline{q}_f$; (5) the standard deviations of the forecast-minus-analysis difference profiles $s_{f-a}$; (6) the number of values in the analysis and the forecast ensemble $N_{a,f}$. These basic variables except the bias of the mean analysis profile are calculated based on statistical calculation using a large ensemble of forecast and analysis profiles in each grid cell. The details of how to extract the ensemble of profiles on the grid and how to calculate these profiles were described by Li et al. [52,53].

The bias profile $b_a$ for temperature is estimated by systematic error modeling according to Li et al. [52,53]. It is applied with no vertical variations but with latitudinal variations. The temperature biases are smallest within the ±40° latitude band, where the values are equal to the basic mean magnitude of $s_0$ (0.5 K). Such values increase with the increase of latitude. Poleward of 60°, $s_0$ are 20% higher than their basic mean magnitude in the summer hemisphere but twice their mean magnitude in the winter hemisphere [51,52]. The bias profile $b_a$ for specific humidity is currently adopted as a relative uncertainty value of 5% of the mean analysis humidity, an educated-guess value.

Using these variables, the background uncertainties $u_b$ (representing both for $u_{Tb}$ and $u_{qb}$) are estimated as:

$$u_b(z) = \left[(u_a(z))^2 + (s_{f-a}(z))^2\right]^{1/2},$$

(33)

where $u_a$ and $s_{f-a}$ here represent the collocated values obtained from a bi-linear interpolation of their values from the four grid points surrounding the tangent-point location of the given RO event. Preparing for this, $u_a$ at each grid point (denoted for clarity as $u_{a\_grid}$) is estimated as a combination of the systematic biases and the statistical errors [52,53]:

$$u_{a\_grid} = \left[b_a^2 + \left(s_a^2/N_{a,f}\right)\right]^{1/2}.$$

(34)

Since background temperature uncertainty $u_{Tb}$ between 10 km and 16 km needs to be penalized to gradually increase in uncertainty at these high tropospheric altitudes, in order to ensure that the observations always safely take increasing weight towards the stratosphere, we modified $u_{Tb}$ from 10 km to 16 km and used an intentional uncertainty increase of the form:

$$u_{Tb}(z) = u_{Tb}\left(z_{Ttop}\right) \cdot e^{\frac{z-z_{Ttop}}{H_{Tb}}},$$

(35)

where $z_{Ttop}$ is 10 km and $H_{Tb}$ is the "uncertainty scale height" set to 5 km.

Background specific humidity uncertainty $u_{qb}$ is input in form of relative humidity values into the scheme. That is, we first use the collocated background specific humidity uncertainty divided by the collocated mean forecast humidity profile, $u_{qb}/\overline{q}_b$, and then use this relative value to multiply it with the collocated actual background profile in order to obtain the specific humidity uncertainty in absolute values for the algorithm.

Figure 2 shows the comparison between OPSv5.6 and the new (dynamic) observation and background uncertainties. It can be seen that $u_{Td}$ increases with decreasing altitude from 0.7 K at 16 km to more than 4 K at the surface. Dry pressure uncertainty stays around 0.2% from 16 km to 10 km and then gradually increases with decreasing altitude to about 1.5% at the surface. As noted above, the observation uncertainties are still used as global static profiles, i.e., used globally in the same way, while in the future rOPS they will be as well used dynamically such as the dynamic background uncertainties discussed next.

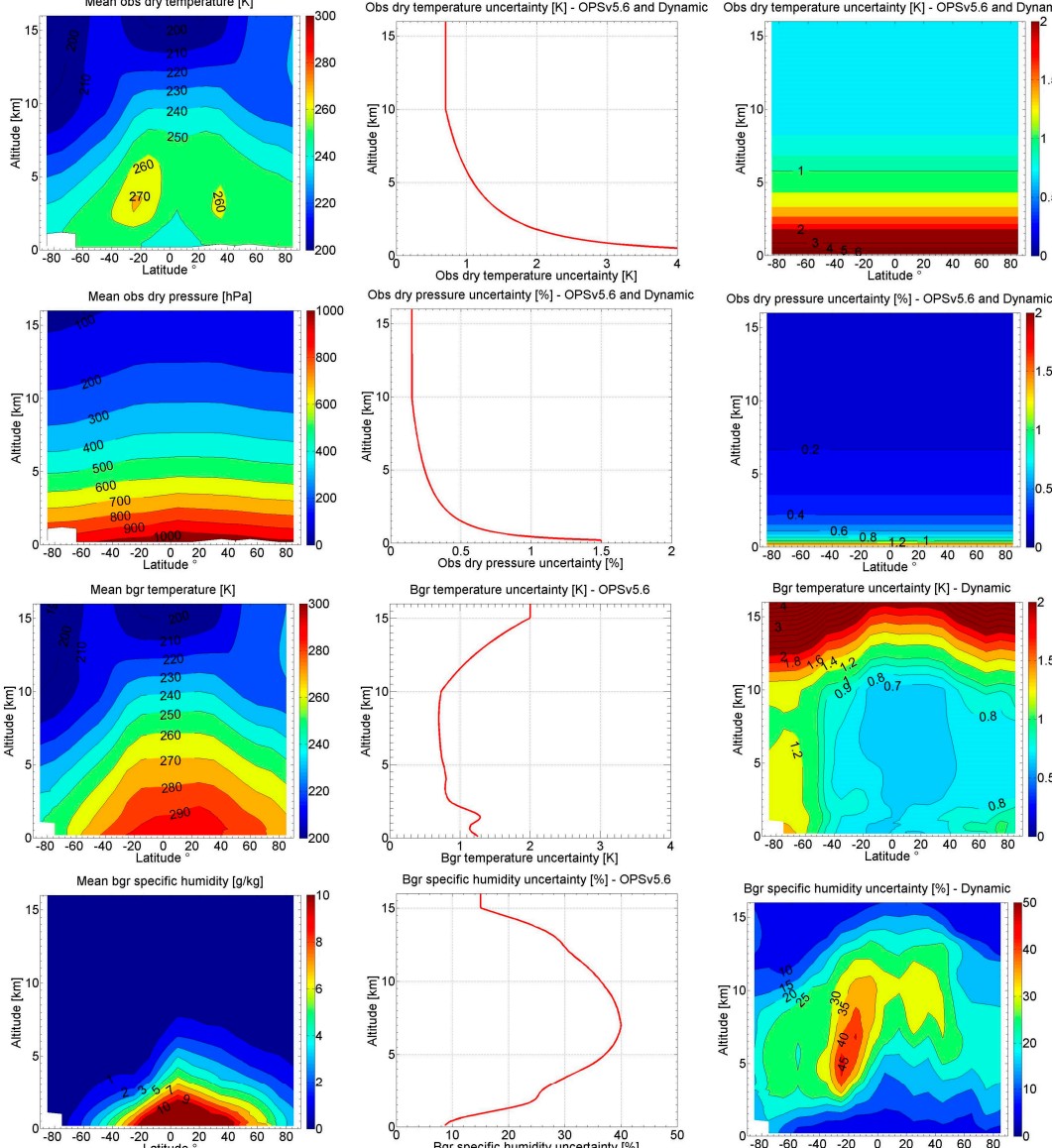

**Figure 2.** Mean profiles and uncertainties of the four input variables, i.e., observed dry temperature (first row), observed dry pressure (second row), background temperature (third row), and background specific humidity (fourth row) on 15 July 2008. For the input observed profiles (first row and second row), which the OPSv5.6 and the dynamic approach share the same uncertainties, the left, middle, and right panels show for the observed mean profile, uncertainty profile as a function of altitude, and the uncertainty profile as a function of altitude and latitude, respectively. For the background profiles (third and fourth row), the left, middle, and right panels show for the background mean profile as a function of altitude, global mean static uncertainty profile of OPSv5.6 approach as a function of altitude, and dynamic background uncertainty as a function of altitude and latitude, respectively.

The third row of Figure 2 shows that the OPSv5.6 $u_{Tb}$ is a static global profile, being 2 K at 16 km, 0.6 K at 10 km, and about 1.2 K at the surface. In comparison, the dynamic $u_{Tb}$ exhibits latitudinal and altitudinal variations. $u_{Tb}$ in polar regions is larger than that in non-polar regions. It is largest in the southern hemisphere polar regions, with values varying from 1.2 K close to the surface to more than 4 K above 12 km. In non-polar regions, the values gradually decrease from high latitudinal regions to low latitudinal regions. Furthermore, our sensitivity test results (not shown) indicate that the dynamic $u_{Tb}$ exhibits clear seasonal variations, with largest uncertainty in the polar winter hemisphere.

The fourth row of Figure 2 shows that relative values of OPSv5.6 uncertainty of background specific humidity $u_{qb}$. It is 10% at the surface, increases to about 40% at 7 km, and then gradually decreases to about 15% at 16 km. The dynamic $u_{qb}$ exhibits clear latitudinal variations, with largest values (>40% from 3 km up to 10 km) in tropical regions, decreasing towards the poles.

We also correct the potential biases that exist in background profiles. This is done by subtracting a background bias profile estimated by using gridded mean forecast profiles minus analysis profiles. It is found that bias-corrected background profiles are useful for getting optimal profiles. For conciseness of the main text of this paper, Appendix B provides more details. Furthermore, we also inspected the vertical correlation structure of background and observation inputs, obtained by constructing error covariance matrices of forecast/observed minus analysis profiles. It is found that the correlations of both background and observation profiles are reasonably small. We hence disregarded to account for these error correlations in the algorithm as introduced in this study; here Appendix C provides more details.

## 2.4. Inspection of Intermediate Variables

In order to provide insight on the characteristics of sub-step results, we illustrate here the input, intermediate, and output variables of the new dynamic approach, using one simulated MetOp (simMetOp) event and one real COSMIC event as representative examples. The MetOp event is simulated under moderate ionospheric conditions. The observational errors represent MetOp/GRAS-type receiving system errors (precise orbit determination (POD) errors, receiver thermal noise, local multipath, clock instabilities), following the proven setting by Steiner and Kirchengast [31], also recently used by Schweitzer et al. [10] and Li et al. [52,53]. The results are shown in Figures 3 and 4.

First focusing on the top row (temperature), it can be seen that $u_{Tb}$ of the simMetOp event, which is located at higher latitudinal regions, is larger than that of the COSMIC event, with values that decrease from 3 K at 16 km to 1 K at 10 km and further to 0.8 K below. For both events, $u_{Tq}$ is smaller than $u_{Tb}$ above 10 km, while below 10 km, $u_{Tq}$ increases quickly and becomes larger than 6 K at bottom altitude levels. The optimally estimated $T_e$ is bounded between $T_b$ and $T_q$ and properly takes more weight from the profile that has ascribed less uncertainty. The differences between $T_e$ and the corresponding reference profile are generally smaller than the differences of $T_b$ and $T_q$, indicating the effectiveness of the optimal estimation. Comparing dynamic $u_{Tb}$ and OPSv5.6 uncertainty $u_{TdOPSv56}$, we can see that $u_{Tb}$ is of similar magnitude as $u_{TdOPSv56}$, with values larger at high latitudes and smaller at low latitudes.

Next focusing on the middle row (specific humidity), and first comparing $u_{qb}$ and $u_{qT}$, we see that $u_{qb}$ is larger than $u_{qT}$ below 8 km for the simMetOp event and below 7 km for COSMIC event. Above 7 km to 8 km, $u_{qT}$ increases quickly to large values. In the optimal estimation, $q_e$ takes more weight from the profile with smaller uncertainties in the optimal estimation, and its difference against the reference profile is smaller than the one between $q_b$ and $q_T$, again indicating the effectiveness of the optimization. The OPSv5.6 specific humidity uncertainty $u_{qbOPSv56}$ is a static profile globally, starting from near 20% at bottom altitude levels, increasing to about 40% at 7 km and then gradually decreasing to below 20% at 16 km.

Finally focusing on the bottom row of Figures 3 and 4 (pressure), the resulting profiles and also the uncertainties are seen basically very close to (or even identical to) each other, due to the dominating factor being the input dry pressure profile and its uncertainty (cf. Equations (A2), (A6), (A8), (A13) in Appendix A and Equations (26) and (27) above). This makes transparent that the pressure chain of computations is quite simple in terms of uncertainty setup in this version, and quite robust in terms of

co-estimating the pressure together with temperature and humidity in a manner consistent with the hydrostatic equation and the equation of state.

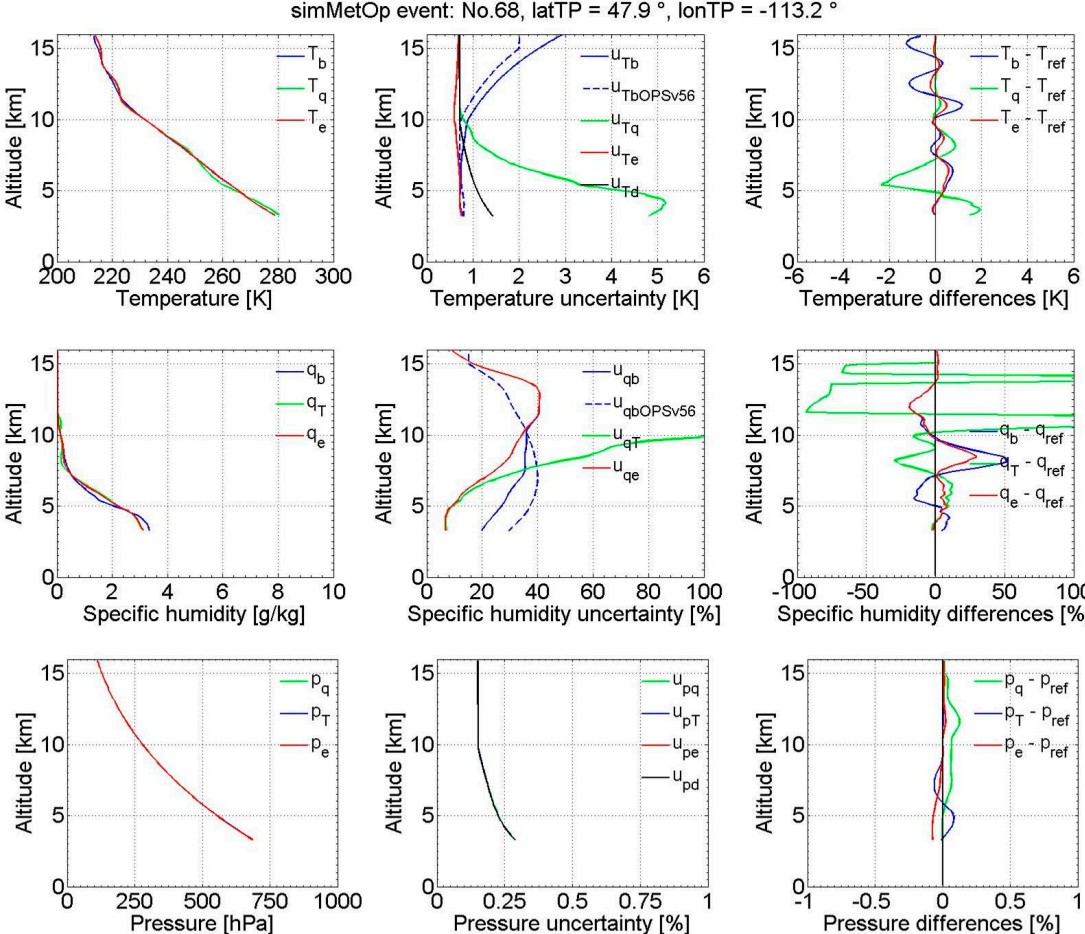

**Figure 3.** Illustration of input, intermediate, and result variables relevant towards the estimation of optimal temperature (top row), specific humidity (middle row), and pressure (bottom row) of an exemplary simMetOp event (identified on top), for the dynamic approach. In the top row, the left panel shows temperature profiles for background temperature (blue), temperature calculated with specific humidity prescribed (green), and the optimally estimated temperature (red). The middle panel shows the estimated uncertainty profiles for the three profiles shown left and for the observed dry temperature (black) as well as the uncertainty of the background temperature from the OPSv5.6 approach (dashed blue). The right panel shows the differences between the temperature profiles shown left and the reference profiles, where the references profiles are the ECMWF co-located analysis profiles. In the three panels of the middle row, the same type of variables is shown as in the upper row, but for specific humidity $q$; thus the intermediate variable here is specific humidity with temperature prescribed (subscript "T") and there is no dedicated input uncertainty profile in the middle panel (such as $u_{Td}$ in the upper row). Similarly, the bottom row shows the corresponding variables for pressure, whereby here the intermediate pressures from both humidity prescribed (subscript "q") and temperature prescribed (subscript "T") are shown together with the optimally estimated pressure (subscript "e"), and the middle panel also illustrates the input uncertainty profile of the dry pressure $p_d$.

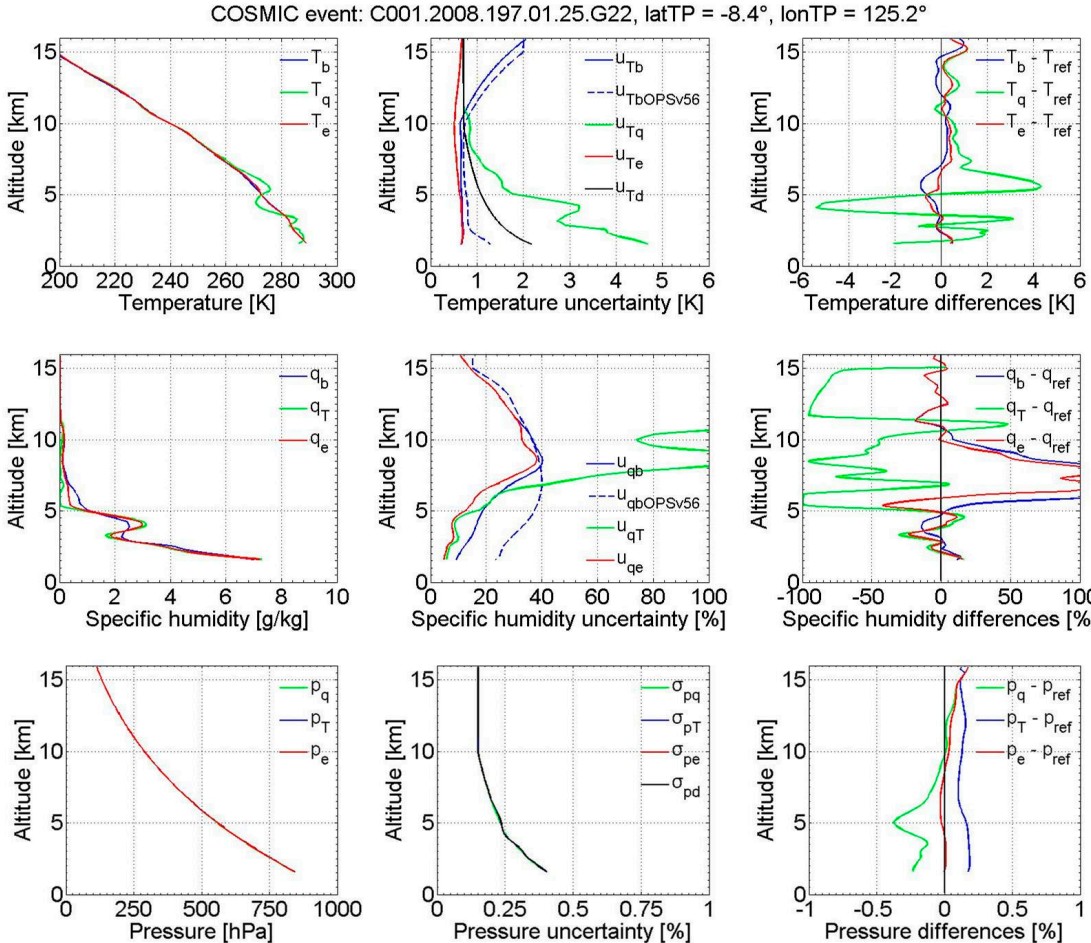

**Figure 4.** Illustration of input, intermediate, and result variables relevant towards the estimation of optimal temperature (top row), specific humidity (middle row), and pressure (bottom row) of an exemplary COSMIC event (identified on top), for the dynamic approach. Figure format and style are the same as for Figure 3; see that caption for explanation.

## 3. Results—Algorithm Performance Evaluation by Comparison to 1DVar Retrievals

The new algorithm in terms of both the OPSv5.6 and dynamic approach is evaluated using simulated and real observed RO data. In addition to these two approaches, moist-air profiles retrieved at ROM-SAF and CDAAC with 1DVar approaches are used for our comparison. The basis of the comparison is to calculate difference profiles between RO retrieved profiles and their corresponding reference profiles based on which we compare the approaches, first by inspecting individual RO events and second by intercomparing statistical profile ensemble results, including systematic differences and standard deviations in different latitudinal bands.

We focus on the comparison of the retrieved profiles of temperature, specific humidity, and pressure. Other moist-air profiles such as density or water vapor pressure profiles, which can be readily derived using these three variables (see Figure 1 and related description), are found to show similar comparative characteristics and are hence for conciseness not additionally discussed here.

Following the successful basic performance evaluation approach of Li et al. [52,53], the data used for the evaluation include simMetOp and real CHAMP and COSMIC data on 15 July 2008, plus, due to the number of CHAMP RO events from one day being not sufficiently high, also CHAMP data on 14 and 16 July 2008. Co-located ECMWF operational analysis profiles are used as reference profiles for all simulated and real RO events.

The End-to-End GNSS Occultation Performance Simulation and Processing System (EGOPS) version v5.6 [54,55] was used for the forward simulation and retrieval of simMetOp data as well as for

the retrieval of the CHAMP and COSMIC profiles by the OPSv5.6 and dynamic approaches. The EGOPS software was developed by WEGC for the simulation of occultation observations and also retrieval, based on the simulated or real observed RO observations. Its retrieval subsystem for RO atmospheric profiles is also independently denoted as OPS. The version 5.6 is its most state-of-the-art version. In simulation of occultation observations, users can carefully select observational noise modeling options, antenna types, orbit accuracy settings, and many other choices. Details on the EGOPS/OPSv5.6 simulation and retrieval capabilities can be found in Fritzer et al. [54] and Schwärz et al. [55].

### 3.1. Insights from Individual Event Profiles

Figure 5 shows the differences between RO retrieved moist profiles and their corresponding reference profiles for three exemplary RO events from simMetOp, CHAMP, and COSMIC for the four approaches evaluated. The three events are intentionally selected to represent at the same time a diversity of latitudes and hence atmospheric conditions, from northern hemisphere middle latitudes (simMetOp) via southern hemisphere polar (CHAMP) to tropical region (COSMIC).

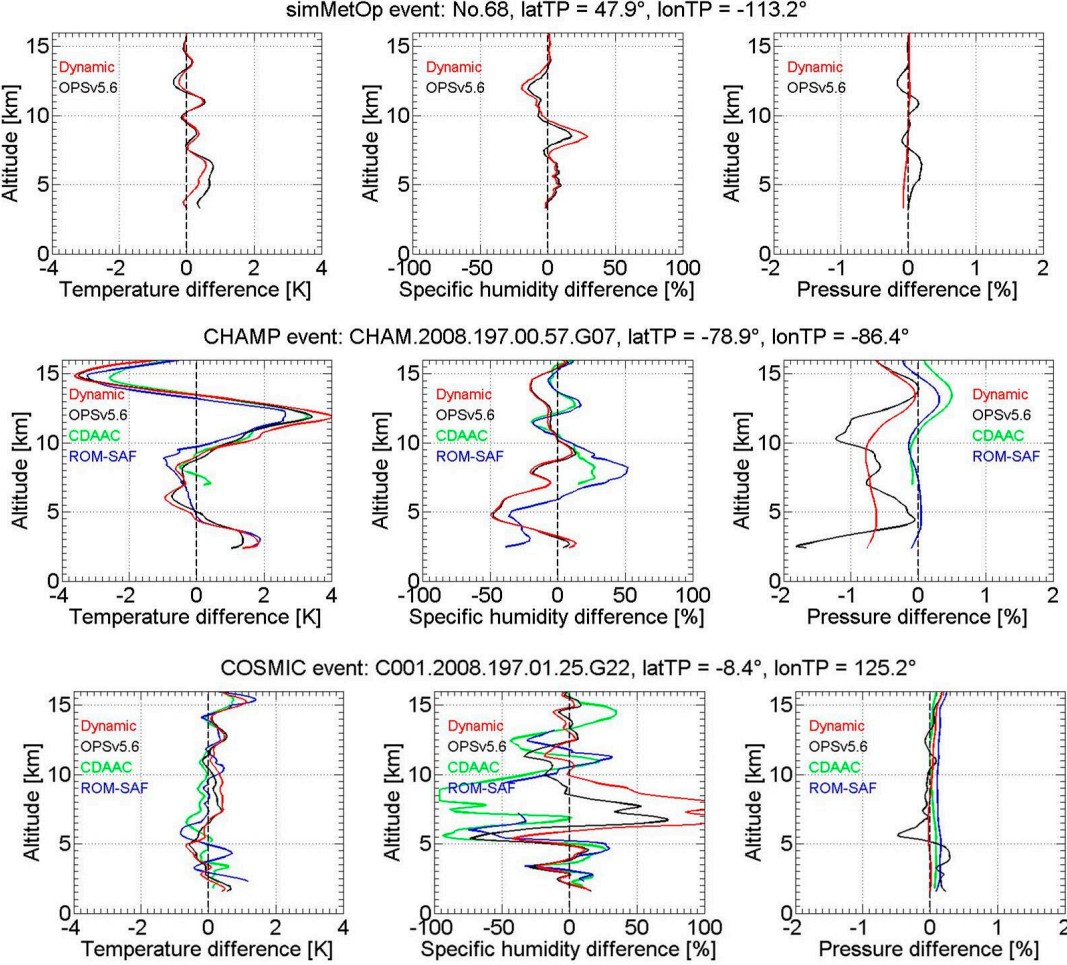

**Figure 5.** Difference profiles between RO-retrieved temperature (left column), specific humidity (middle column), and pressure profiles (right column) and their corresponding ECMWF co-located analysis profiles, for three exemplary events (identified on top of each row) from simMetOp (top row), CHAMP (middle row), and COSMIC (bottom row), respectively. The results for the dynamic (red), OPSv5.6 (black), CDAAC (green), and ROM-SAF (blue) approaches are shown.

From the simMetOp event we can see that the temperature and pressure differences from the dynamic approach are smaller than those of the OPSv5.6 approach, while specific humidity differences

from the dynamic approach are larger than those from OPSv5.6. For the CHAMP event, temperature differences of the four approaches are generally consistent. Specific humidity differences of the four approaches are generally similar, with the OPSv5.6 and dynamic approaches showing somewhat larger differences from about 4 km to 7 km and CDAAC and ROM-SAF larger ones from about 7 km to 10 km.

Pressure differences from OPSv5.6 are largest among the four approaches and exhibit some smaller-scale altitude variations. This is expected according to the algorithm choice (cf. Section 2.2) that optimal pressure in the OPSv5.6 approach is calculated consistent with the Smith-Weintraub formula, rather than with the hydrostatic equation, and is hence affected by the errors/fluctuations of dry temperature. Pressure differences from ROM-SAF are smallest for the CHAMP event and largest for the COSMIC event amongst the four approaches, indicating event-to-event variation in how estimated temperature and humidity play together in yielding pressure profiles.

For the COSMIC event, temperature differences for the four approaches are again rather similar. Specific humidity differences from all four approaches are generally consistent as well, with slightly larger values from the dynamic approach between 5 km to 10 km. While these inspections provide some insights to typical individual-event behavior, a more reliable comparison based on statistical results is needed as discussed below.

## 3.2. Statistical Ensemble Results

In order to investigate the statistical performance of the four approaches in different latitudinal regions, the error statistics in terms of the systematic differences and standard deviations of the retrieved profiles against the reference profiles are calculated in six representative latitudinal regions, comprising global total (90°S to 90°N) and five latitude bands (see Figure 6 and its caption).

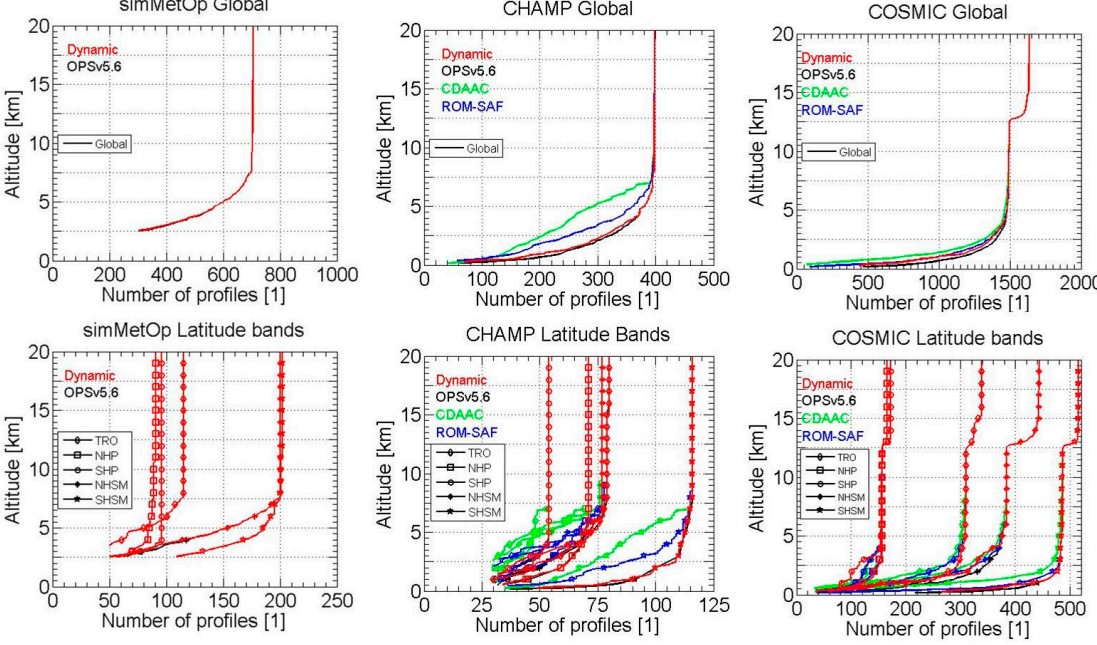

**Figure 6.** Number of RO profiles from simMetOp (left), CHAMP (middle), and COSMIC (right) as function of altitude for the global domain (top row) and five latitudinal bands (bottom row), including TRO (tropics; 20°S to 20°N), NHP (northern hemisphere polar; 60°N to 90°N), SHP (southern hemisphere polar; 60°S to 90°S), NHSM (northern hemisphere subtropics and mid-latitudes, 20°N to 60°N), and SHSM (southern hemisphere subtropics and mid-latitudes, 20°S to 60°S), on 15 July 2008 for simMetOp and COSMIC and on 14-16 July 2008 for CHAMP. The red, black, green, and blue colors denote the dynamic, OPSv5.6, CDAAC, and ROM-SAF approaches, respectively, with the dynamic one plotted last (hence shadowing other colors above the lower to middle troposphere) and the profiles for different latitude bands denoted by distinct symbols (see legend).

In order to avoid outliers and ensure an identical RO event ensemble for all four approaches, RO profiles that showed bad quality in any of the processing systems, based on the quality control settings and flags in the respective data files supplied, were rejected from the joint event ensemble. In particular, the quality of retrieved profiles from the OPSv5.6 and dynamic approaches was determined by the OPSv5.6 system specifications [56] and the quality of the profiles retrieved at CDAAC and ROM-SAF by the system they used at their centers [20,26,57].

Figure 6 shows the resulting number of profiles (i.e., number of RO events) available for the joint statistical evaluation for the four approaches in the six latitudinal bands. Given our joint-events selection noted above, the number of profiles above about 5 km to 8 km altitude is the same for the four approaches; only in the lower and middle troposphere below about 8 km there is a number-of-profiles reduction depending on the specific processing systems and their specific criteria to cut the tropospheric penetration of individual moist-air profiles depending on retrieval quality.

In general, the number of profiles available deep into the troposphere from the OPSv5.6 and dynamic approach is somewhat larger than that from CDAAC and ROM-SAF. Furthermore, the number of ensemble members in the five latitude bands varies from about 50 (CHAMP in SHP) to about 500 profiles (COSMIC in SHSM), with all bands enabling reasonable statistics for this initial comparative performance evaluation study among the four approaches.

Figures 7–9 illustrate the statistical results for the RO retrieved profiles of simMetOp, CHAMP, and COSMIC, respectively. Figure 7 shows for the simMetOp profiles ensemble that the systematic differences for temperature and specific humidity from the dynamic approach are smaller than those from the OPSv5.6 approach. The best relative improvements are found in tropical regions, where the temperature systematic differences of the dynamic approach are 0.2 K smaller than those of the OPSv5.6 approach, and specific humidity differences are about 15% smaller.

Standard deviations of temperature and specific humidity from the dynamic approach are smaller than or similar to those of OPSv5.6. The propagated uncertainties of temperature, $u_{Te}$ are larger than the statistically estimated standard deviations, which is especially related to the fact that $u_{Te}$ is calculated using $u_{Td}$ (cf. Equations (14) and (21)), which is empirically estimated for real rather than simulated data based on the model by Scherllin-Pirscher et al. [51]. That is, for simulated data, $u_{Td}$ is overestimated since the quality of dry temperature of our simulated data is better than real observed data. The propagated specific humidity uncertainties are of similar magnitude compared to the statistically estimated uncertainties.

The statistical differences of pressure for the dynamic approach are clearly smaller than those from OPSv5.6, which is due to the different closure-scheme of the pressure computation as discussed above (Section 2.2). The propagated pressure uncertainties are larger than the statistically estimated uncertainties, similar to temperature, which is similarly related to the overestimation of the uncertainty of dry pressure, targeted to real data, for these simulated MetOp data.

Figure 8 shows for the CHAMP profiles ensemble that the temperature error statistics in terms of systematic differences and standard deviations from the dynamic, OPSv5.6, and ROM-SAF approaches are rather similar in all latitudinal bands, with systematic differences reaching around ±0.2 K and standard deviations being smaller than 1 K down to the boundary layer. Temperature statistics of CDAAC are as well rather similar to the other three approaches, with standard deviation only slightly larger below about 8 km. This slightly larger standard deviation of CDAAC is probably due to a somewhat stronger weighting of observations vs. background at low to middle troposphere levels, where observations are nosier compared to background. Furthermore, it needs to be kept in mind that the reference profiles are for mean RO event locations, while actual tangent points drift during occultation (e.g., [19]), which also contributes to enlarged deviations at lowest tropospheric levels. The propagated temperature uncertainties $u_{Te}$ are basically consistent with the statistically estimated uncertainty, which again indicates the reasonableness of this simplified uncertainty propagation.

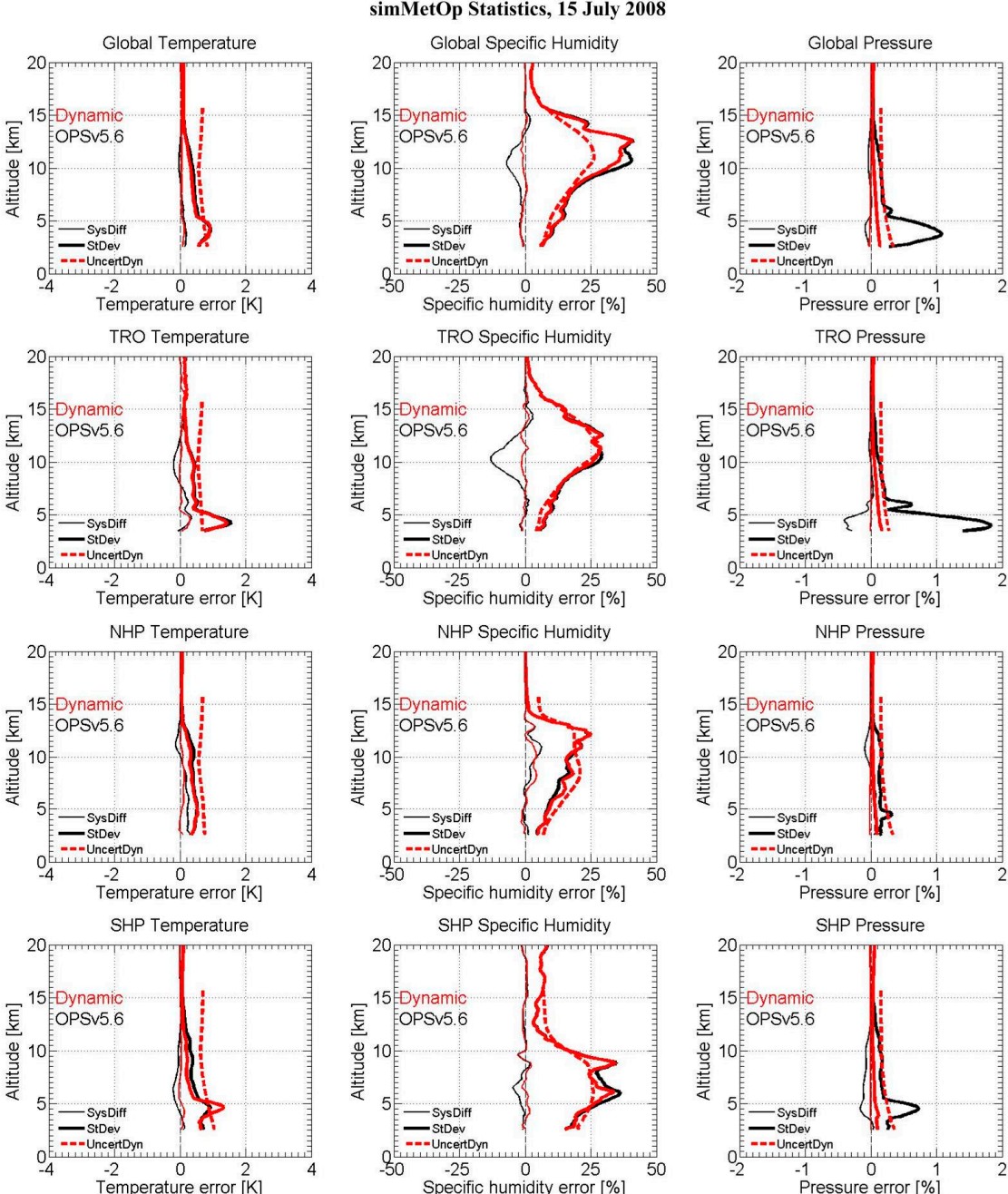

**Figure 7.** Systematic differences (SysDiff) and standard deviations (StDev) of retrieved temperature (left column), specific humidity (middle column), and pressure (right column), relative to ECMWF co-located analysis profiles as reference, of the ensemble of simMetOp events on 15 July 2008. Statistics for both the dynamic (red) and OPSv5.6 (black) approach are shown for four representative regions (top to bottom: Global, TRO, NHP, SHP). The propagated uncertainties of retrieved profiles from the dynamic approach (UncertDyn; red-dashed) are shown as well.

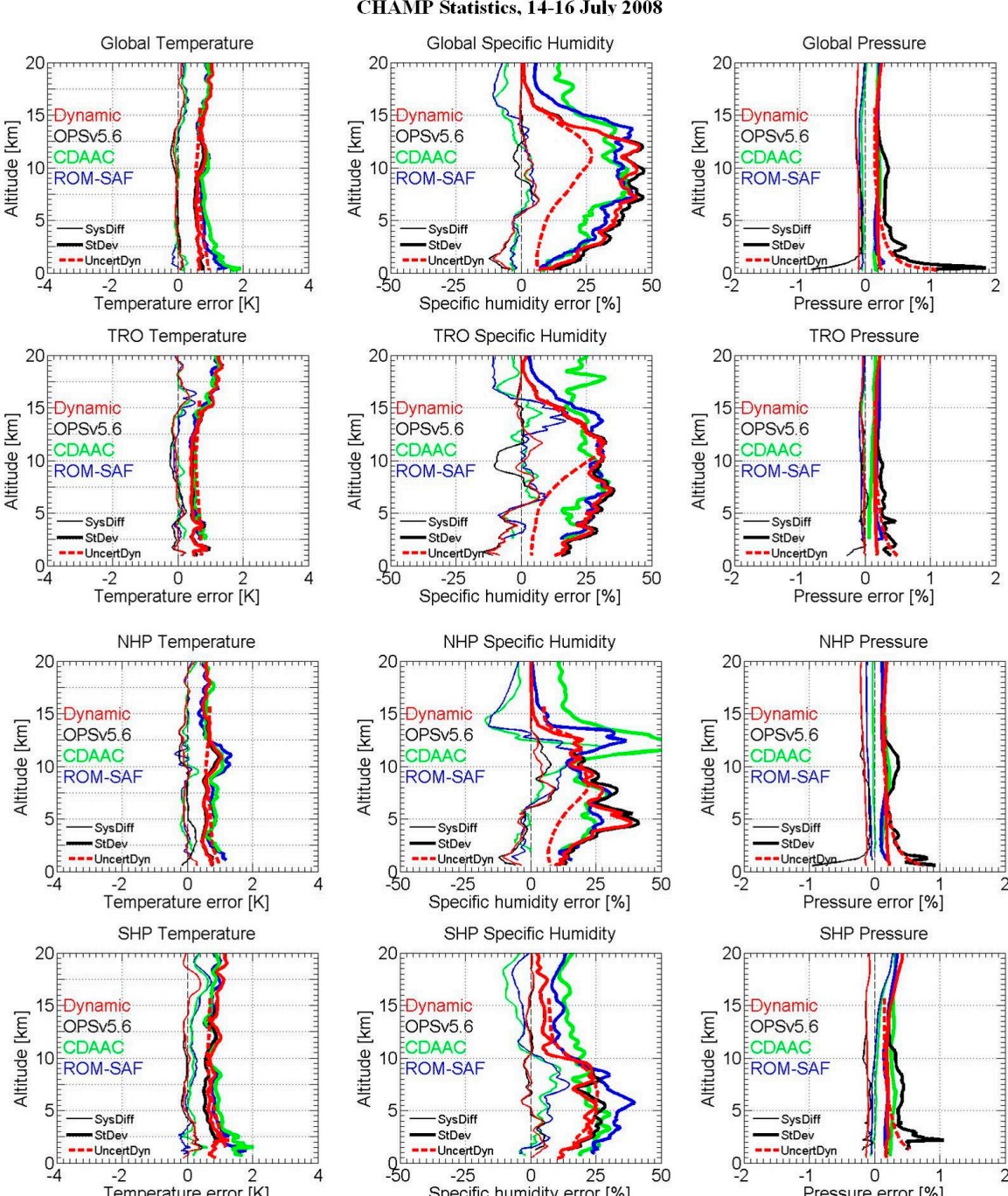

**Figure 8.** Systematic differences (SysDiff) and standard deviations (StDev) of retrieved temperature (left column), specific humidity (middle column), and pressure (right column), relative to ECMWF co-located analysis profiles as reference, of the ensemble of CHAMP events on 14-16 July 2008. Statistics for the dynamic (red), OPSv5.6 (black), CDAAC (green), and ROM-SAF (blue) approach are shown for four representative regions (top to bottom: Global, TRO, NHP, SHP). The propagated uncertainties of retrieved profiles from the dynamic approach (UncertDyn; red-dashed) are shown as well.

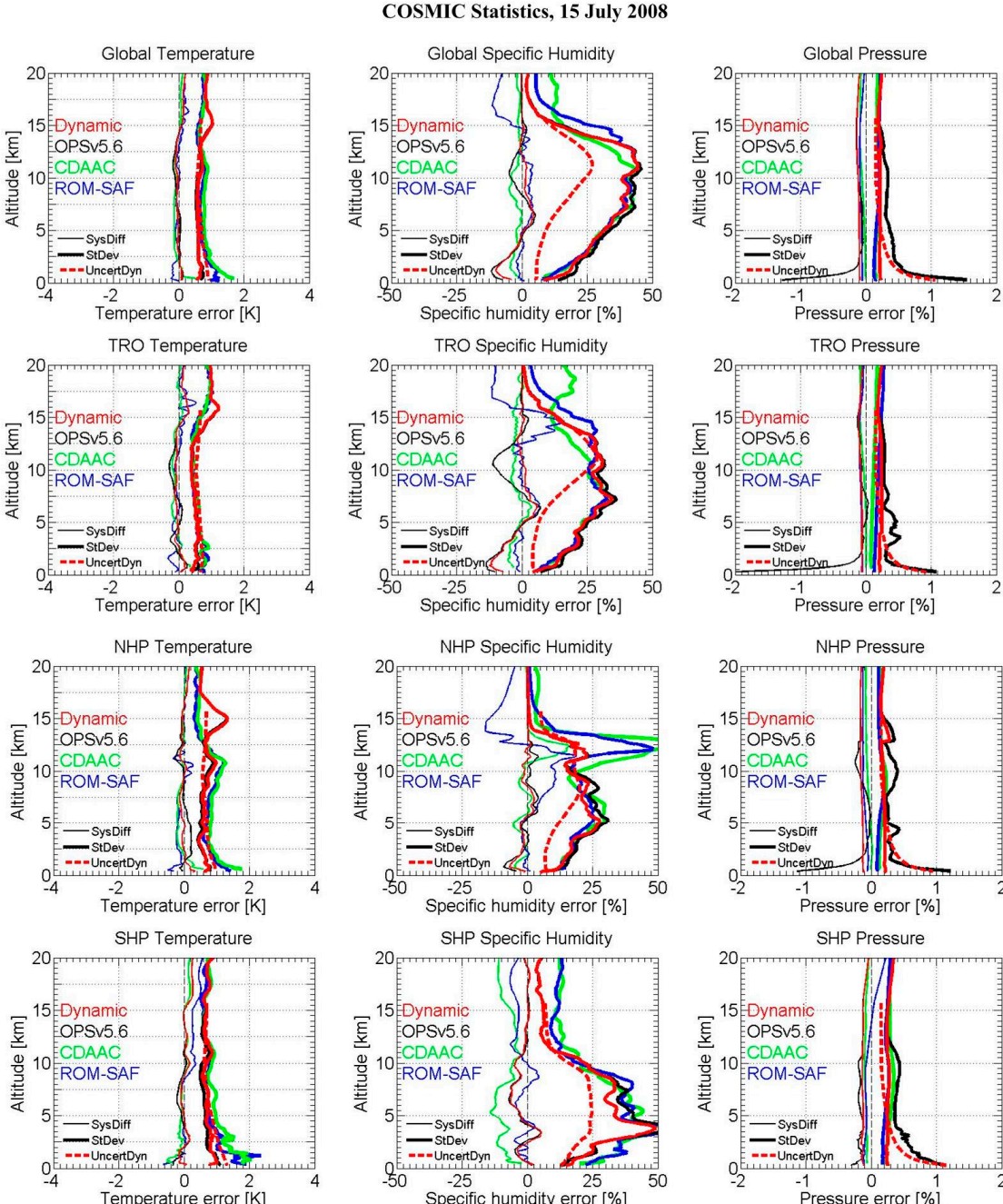

**Figure 9.** Systematic differences (SysDiff) and standard deviations (StDev) of retrieved temperature (left column), specific humidity (middle column), and pressure (right column), relative to ECMWF co-located analysis profiles as reference, of the ensemble of COSMIC events on 15 July 2008. Statistics for the dynamic (red), OPSv5.6 (black), CDAAC (green), and ROM-SAF (blue) approach are shown for four representative regions (top to bottom: Global, TRO, NHP, SHP). The propagated uncertainties of retrieved profiles from the dynamic approach (UncertDyn; red-dashed) are shown as well.

The specific humidity error statistics from the four approaches are basically consistent, on a global-mean scale, with systematic differences reaching around ±10% and standard deviations varying from 10% to 50%. In more detail, the specific humidity statistics reveal clear altitudinal variations, with largest values in the upper troposphere (about 5 km to 10 km), and also latitudinal variations. For example, in the NHP region, error statistics of ROM-SAF and CDAAC are found to be comparatively larger above about 10 km, which is possibly related to the treatment of very small specific humidity

values, where WEGC (dynamic, OPSv5.6) places a stronger constraint towards the background. Therefore, WEGC profiles have less deviation from the (ECMWF-based) background humidity and also from the ECMWF analysis reference here.

The propagated uncertainty of the specific humidity, $u_{q\mathrm{e}}$, is generally smaller below about 10 km than the statistically estimated uncertainties (except in the SHP region, where absolute moisture content is low). This is due to the reason that observed, background, and analysis specific humidity jointly represent more (noisy) variations over the troposphere than the simplified error estimates used here do capture. In other words, part of the "representativeness error" is not captured.

Pressure error statistics below 10 km from all four approaches are basically rather consistent, with systematic differences varying within about ±0.1% and standard deviations reaching about ±0.2%. Pressure differences of the OPSv5.6 approach are somewhat larger and exhibit comparatively more fluctuations below 5 km (below 2 km for systematic differences), which is due to the different refractivity-based (rather than hydrostatic-based) closure scheme explained in the algorithm description above (Section 2.2).

The propagated uncertainties of pressure are similar to the statistically estimated uncertainties above about 5 km. Below 5 km, the propagated uncertainties are found somewhat larger than the statistically estimated uncertainty. This is due to the reason that the simplified input uncertainty estimate used here for dry pressure, which is the basis for calculating the propagated uncertainty, is rather large at low altitudes. In WEGC's future rOPS context, this input uncertainty will be more realistic as part of a full chain of uncertainty propagation [44].

Figure 9 shows for the COSMIC profiles that the comparative performances of the four approaches are similar to what is visible and was just discussed for the CHAMP profiles shown in Figure 8. We note that in the tropical lower troposphere below about 3–4 km, where moisture content is largest, the OPSv5.6 and dynamic approaches exhibit a negative systematic difference of up to around −10% in specific humidity. Similar to other differences between the approaches visible in the upper troposphere, this is likely related to different uncertainty weighting choices and points to room for further refinement in future.

### 3.3. Simple Observation-to-Background Weighting Ratio Profiles and Comparative Results

In order to know how much observation information was used in the retrieved moist profiles, we calculate observation-to-background uncertainty weighting ratio profiles, $r_{\mathrm{obw}}$, of temperature and specific humidity for the dynamic, OPSv5.6 and ROM-SAF approaches (CDAAC-provided moist profile files do not contain uncertainty information, hence these data are not included here). Since both our approaches and the 1DVar approach used by ROM-SAF are not (fully) linear, it is not an easy task to calculate the real $r_{\mathrm{obw}}$ in a comparable manner. Hence we implemented and inspected an approximation as follows.

Considering that the calculation of retrieval-to-background uncertainty ratio ($r_{\mathrm{rbu}}$) is straightforward and consistently possible for all four datasets, we used this ratio to calculate an approximate $r_{\mathrm{obw}}$. The retrieval-to-background uncertainty ratio is defined and calculated as $r_{\mathrm{rbu}} = 100 \frac{u_{\mathrm{ret}}}{u_{\mathrm{b}}}$, where $u_{\mathrm{ret}}$ is the uncertainty profile of the retrieved (optimally estimated) profile and $u_{\mathrm{b}}$ is the corresponding background uncertainty profile. Based on the $r_{\mathrm{rbu}}$ profile, we can then estimate $r_{\mathrm{obw}}$ as:

$$r_{\mathrm{obw}} = 100\left(1 - \left(\frac{r_{\mathrm{rbu}}}{100}\right)^2\right) = 100\left(1 - \frac{u_{\mathrm{ret}}^2}{u_{\mathrm{b}}^2}\right), \tag{36}$$

which implicitly assumes an inverse-variance weighted combination of observations and background in the optimal estimation, being a reasonable approximation.

Figure 10, illustrating the $r_{\mathrm{obw}}$ results for temperature and specific humidity, shows that the $r_{\mathrm{obw}}$ for temperature from dynamic, OPSv5.6, and ROM-SAF approaches are generally consistent, with dynamic $r_{\mathrm{obw}}$ comparatively largest. This indicates that, by its observational and background

uncertainty choices, the dynamic approach uses more observation information in the temperature retrieval. The temperature results of the three approaches reveal clear latitudinal variations, with less observation information used in the tropics (TRO), where humidity is large, and more in polar regions (especially SHP), where humidity is small and RO likewise accurate.

**Figure 10.** Observation-to-background weighting ratio profiles for temperature (six panels in upper two rows) and specific humidity (six panels in lower two rows), for the COSMIC data ensemble of 15 July 2008, are shown for the global ensemble (Global) and the five latitudinal bands TRO, SHSM, NHSM, SHP, and NHP (identified in the panel titles). The results for the dynamic, OPSv5.6 and ROM-SAF approaches are all shown.

The $r_{\mathrm{obw}}$ profiles of specific humidity from all three approaches are generally consistent, with values from ROM-SAF approach largest (except for NHSM and NHP region), in line with how the temperature weighting is tentatively going the other way. That is, higher relative weight on temperature will generally lead to lower relative weight on humidity, and vice versa. In NHSM and NHP region, values from OPSv5.6 are largest probably due to larger background errors (cf. Figure 2) and subsequent more observation information used in the final optimal estimation.

## 4. Discussion

The results of Section 3 and Figure 5, Figure 8, and Figure 9 provide clear evidence that our new "simplified 1DVar" approach with the "direct-retrieval method" results obtained as intermediate step, both in form of the OPSv5.6 and the dynamic approaches, provide (at least) the same level of quality as the ROM-SAF and CDAAC "full 1DVar" approaches. Especially when comparing the results from the OPSv5.6 and ROM-SAF approaches, which use the most similar background uncertainties, the error statistics of these two approaches are generally very close.

In general, the background and observation uncertainties are key for determining the weights of background and observations. Hence mainly the weights of background and observations in the optimal estimation determine the statistical errors of moist profiles, which depend on RO data processing centers' evaluation of the quality of background and observation data. Comparing the OPSv5.6 and the dynamic approach, we find that the dynamic approach has reduced the systematic differences of temperature and the statistical errors of humidity as well as improved the thermodynamic consistency of the pressure results, which confirms the effectiveness of improvements of the dynamic approach on top of the OPSv5.6 approach.

The inter-comparison results here demonstrate the performance and general accuracy of the new algorithm in its basic form, without accounting for error correlations and further uncertainty propagation advancements. Schwarz [44] advanced the dynamic approach by propagating estimated random uncertainties using covariance propagation, controlled by Monte-Carlo ensemble methods. The covariance propagation, accounting for error correlations, also enabled to implement a full covariance-weighted optimal estimation. These specific most recent advancements are published elsewhere, together with a further step of performance evaluation on its added value.

Overall it is already clear from the results of this study that the "simplified 1DVar", with its special features of step-by-step transparency of state retrieval as well as systematic and random uncertainty propagation, is a viable new algorithm achieving the quality of "full 1DVar".

## 5. Conclusions

In this study, a new sequentially linearized "simplified 1DVar" algorithm was introduced that combines the so-called direct method, with temperature or humidity prescribed, with optimal estimation, for providing accurate temperature, humidity, and pressure profiles from RO in the troposphere. It was also evaluated using the "full 1DVar" algorithm implementations from the ROM-SAF and CDAAC processing centers.

While approximating the matrix inversion and iteration approach used in 1DVar algorithms in simplified form, we find the new algorithm to nevertheless effectively allow retrieving accurate optimally estimated profiles, along with systematic and random uncertainty propagation and effective observation-to-background weighting ratio tracking. The direct-method retrieval results, temperature profiles with background humidity profiles prescribed as well as humidity profiles with background temperature profiles prescribed, are available as intermediate results and can hence be considered a useful by-product.

The uncertainties of background and observational variables are dynamically estimated in the new algorithm, using statistical calculations and empirical modeling. The estimated uncertainties account for latitudinal and seasonal variations. Residual biases in background profiles (ECMWF short-range forecast profiles) are corrected for by using co-located ECMWF forecast-minus-analysis difference bias profiles and are found useful in reducing biases in resulting profiles.

The comparison of the new algorithm against the moist-air profiles provided by the current OPSv5.6 processing system and profiles from ROM-SAF and CDAAC showed that it provides robust and high quality temperature, humidity, and pressure profiles in the troposphere, comparable in performance with "full 1DVar", plus uncertainty estimates in good quality.

The new algorithm was implemented in the OPSv5.6 system with static uncertainty profiles as an initial scope, while the further advanced dynamic approach presented in this paper, is using dynamic uncertainties and the further improvements described. In future, the algorithm in a further advanced form, based on the work by Schwarz (2018) [44], will be used as part of the WEGC's new rOPS processing system. This rOPS-implemented moist-air algorithm that is built on the algorithm introduced in this study, is also used in the first large-scale reprocessing towards a tropospheric climate data record 2001-2019 by the rOPS and its integrated uncertainty propagation.

**Author Contributions:** Conceptualization, Y.L. and G.K.; Data curation, Y.L., B.S.-P., M.S. and J.K.N.; Formal analysis, Y.L., G.K., B.S.-P. and S.-p.H.; Project administration, Y.-b.Y.; Software, M.S. and Y.L.; Supervision, G.K.; Validation, Y.L., J.K.N. and S.-p.H.; Visualization, Y.L. and G.K.; Writing—original draft, Y.L. and G.K.; Writing—review & editing, G.K., Y.L. B.S.-P., J.K.N. and S.-p.H. Y.L. implemented the advancements of the OPSv5.6 by the dynamic algorithm, performed the analysis, produced the figures, and wrote the initial draft of the manuscript. G.K. served as primary coauthor, providing advice and guidance on all aspects of the design, analysis, figure production, and initial drafting of the manuscript and significantly contributed to writing of the text. B.S.-P. contributed to the initial-phase advice and guidance of the manuscript preparation and its initial drafting, provided observational uncertainty data, and commented and edited the manuscript in the completion phase. M.S. supported the setup and advancements of the OPSv5.6 analysis system, advised on data and algorithm implementation aspects, and contributed to the finalization of the manuscript writing. J.K.N. provided the data from ROM SAF, S.-p.H. provided advice in the design and initial drafting phase, and J.K.N., S.-p.H., and Y.-b.Y. all commented on the manuscript in different phases, advised on specific aspects of analysis and interpretation, and contributed to finalization of the writing. The manuscript contents are solely the opinions of the author(s) and do not constitute a statement of policy, decision, or position on behalf of NOAA or the U.S. Government.

**Funding:** This research was funded by the Strategic Priority Research Program of Chinese Academy of Sciences (Grant No. XDA17010304), Chinese Natural Sciences Foundation (grant no. 41504035, 4187404, 41574033). At the WEGC Graz side, this work was funded by the Austrian Research Promotion Agency (FFG) projects OPSCLIMTRACE and OPSCLIMVALUE, and the Austrian Science Fund (FWF) project DYNOCC (grant no. T620-N29). J. K. Nielsen has been supported by the Radio Occultation Meteorology Satellite Application Facility (ROM SAF), which is a decentralized operational RO processing center under EUMETSAT.

**Acknowledgments:** We thank the UCAR COSMIC Data Analysis and Archiving Center (CDAAC) for providing access to their RO atmospheric profiling data and Tae-Kwon Wee (UCAR) for valuable discussions. Furthermore, we thank the ECMWF (Reading, UK) for providing access to their analysis and forecast data, and the RO software development and data processing teams at WEGC and ROM SAF for their support in the processing and provision of the RO data from their center used in the study.

**Conflicts of Interest:** The authors declare no conflict of interest.

## Appendix A  Detailed Numerical-Algorithm Formulations of Steps 1a and 1b

This Appendix describes the detail numerical-integration formulation of step 1a "retrieval of temperature and its uncertainty with specific humidity prescribed" and step 1b "retrieval of specific humidity and its uncertainty with temperature prescribed". Based on this description, interested readers should be enabled to implement this approach also in their processing systems. The description provides details of practical implementation expertise at WEGC, such as on assigning robust initial values and ensuring very rapid convergence of iterations, which may help save substantial testing and tuning time in case of re-implementation in other systems.

*Step 1a–Retrieval of temperature and its uncertainty with specific humidity prescribed*

The inputs of this retrieval step include the prescribed background specific humidity $q_b$ and its associated uncertainty $u_{qb}$, the observed dry temperature $T_d$ and its uncertainty $u_{Td}$, and the observed dry pressure $p_d$ and its uncertainty $u_{pd}$. As noted in the main text of the paper, $V_{wb}$ can be calculated using Equation (3). Then, based on Equations (12) and (13), the profiles $T_q(z)$ and $p_q(z)$ can be solved by iteration, level by level top-downward from the level below the first level ($z_{\text{iniMoist}} = 16$ km) to the bottom level, of the ($T$-$\beta$-$p$)-three-equation system:

$$T_{q,k+1}(z_i) = T_{\mathrm{d}}(z_i)\frac{p_{q,k}(z_i)}{p_{\mathrm{d}}(z_i)}\left(1 + \frac{c_T}{T_{q,k}(z_i)}V_{w\mathrm{b}}(z_i)\right), \tag{A1}$$

$$p_{q,k+1}(z_i) = p_q(z_{i-1})\left(\frac{p_{\mathrm{d}}(z_i)}{p_{\mathrm{d}}(z_{i-1})}\right)^{\beta_{q,k+1}(z_{i-1/2})}, \tag{A2}$$

where $\beta_{q,k+1}(z_{i-1/2}) = \frac{T_{\mathrm{d}}(z_i)+T_{\mathrm{d}}(z_{i-1})}{T_{q,k+1}(z_i)+T_q(z_{i-1})}\cdot\frac{1+b_w\sqrt{V_{w\mathrm{b}}(z_i)V_{w\mathrm{b}}(z_{i-1})}}{1+2b_w\sqrt{V_{w\mathrm{b}}(z_i)V_{w\mathrm{b}}(z_{i-1})}}$.

At each altitude level $z_i$, initial values for the iteration are ($k = 0$):

(1) $T_{q,0}(z_i) = T_q(z_i) + 0.8\cdot c_{q2T}\cdot q_\mathrm{b}(z_i)$ and $p_{q,0}(z_i) = p_{\mathrm{d}}(z_i) - 0.2\cdot c_{q2T}\cdot q_\mathrm{b}(z_i)p_{\mathrm{d}}(z_i)/T_{\mathrm{d}}(z_i)$, if $T_{\mathrm{d}}(z_i) \leq T_{\mathrm{dThres}} \vee z_i = z_{\mathrm{iniMoist}}$, where $T_{\mathrm{dthres}} = 240$ K is a threshold in dry temperature $T_{\mathrm{d}}$ above which it can typically deviate by more than 1 K from actual $T$;

(2) $T_{q,0}(z_i) = T_q(z_{i-1})$ and $p_{q,0}(z_i) = p_q(z_{i-1})\cdot(1 + |z_i - z_{i-1}|/H_0)$ if $T_{\mathrm{d}}(z_i) > T_{\mathrm{dThres}} \wedge z_i < z_{\mathrm{iniMoist}}$, where $H_0 = 8$ km.

The iteration ends at $k = k + 1$ that satisfies $\left|T_{q,k+1}(z_i) - T_{q,k}(z_i)\right| < dT_{\mathrm{tol}}$, where $dT_{\mathrm{tol}} = 0.01$ K is the convergence tolerance. At all higher levels, $z_i > z_{\mathrm{iniMoist}}$, use the same formulations to assign $T_q$ and $p_q$ as used at $z_{\mathrm{iniMoist}}$, i.e., the initial-value formulations under iteration condition (1) above.

In order to obtain the uncertainty of the retrieved $T_q$, we first derive a linearized version of Equation (12). Using the approximate assumptions of $V_{\mathrm{wb}} \approx q_\mathrm{b}/a_w, dp_d/p_d \approx dp_q/p_q, dT_d/T_d \approx dT_q/T_q$ and $dT_d/T_d \wedge dT_q/T_q << dV_{wb}/V_{wb}$, which are reasonably valid over the moist air retrieval altitude range, the linearized version becomes:

$$dT_q = \left(\frac{p_q}{p_{\mathrm{d}}}\right)dT_d + \left(\frac{p_q}{p_{\mathrm{d}}}\frac{T_{\mathrm{d}}}{T_q}c_{q2T}\right)dq_\mathrm{b}. \tag{A3}$$

Based on this linear relation, the variance profile of retrieved temperature $u_{Tq}^2$ can be calculated using $u_{Td}^2$ and $u_{q\mathrm{b}}^2$ according to the variance-based uncertainty propagation law:

$$u_{Tq}^2 = \left(\frac{p_q(z)}{p_{\mathrm{d}}(z)}\right)^2 u_{Td}^2(z) + \left(\frac{p_q(z)}{p_{\mathrm{d}}(z)}\frac{T_{\mathrm{d}}(z)}{T_q(z)}c_{q2T}\right)^2 u_{q\mathrm{b}}^2(z), \tag{A4}$$

so that the square-root of this result is the uncertainty profile of the retrieved temperature profile with specific humidity specified: $u_{T\mathrm{q}}(z)$.

Similarly, based on Equations (9) and (14), reasonably assuming that $d\ln p(z) \approx dp(z)/p(z)$ and $d\ln p_\mathrm{d}(z) \approx dp_\mathrm{d}(z)/p_\mathrm{d}(z)$, we can write here as linearized version:

$$dp_q(z) = \beta_q'(z)\frac{p_q(z)}{p_\mathrm{d}(z)}dp_\mathrm{d}, \tag{A5}$$

where $\beta_q(z) = \frac{T_\mathrm{d}(z)(1+b_\mathrm{w}V_{\mathrm{wb}}(z))}{T_q(z)(1+2b_\mathrm{w}V_{\mathrm{wb}}(z))}$. Using this single-term result, the uncertainty propagation is straightforward and the uncertainty profile of the retrieved pressure profile with specific humidity prescribed, $u_{p\mathrm{d}}(z)$, is obtained via:

$$u_{pq}(z) = \beta_q(z)\left(\frac{p_q(z)}{p_\mathrm{d}(z)}\right)u_{p\mathrm{d}}(z). \tag{A6}$$

*Step 1b–Retrieval of specific humidity and its uncertainty with temperature prescribed*

The inputs of this step are the prescribed background temperature $T_\mathrm{b}$ and its uncertainty $u_{T\mathrm{b}}$, the observed dry temperature $T_\mathrm{d}$ and its uncertainty $u_{T\mathrm{d}}$, and the observed dry pressure $p_\mathrm{d}$ and its uncertainty $u_{p\mathrm{d}}$. Using these input profiles, we can solve for profiles $V_{\mathrm{w}T}$ and $p_\mathrm{T}$ based on Equations (16)

and (17), based on iterating level by level as for step 1a above. The ($V$-$\beta$-$p$)-three-equation system in this case is:

$$V_{wT,k+1}(z_i) = \frac{\frac{p_d(z_i)}{p_{T,k}(z_i)} T_b(z_i) - T_d(z_i)}{c_T \frac{T_d(z_i)}{T_b(z_i)}} \geq \frac{q_{minE}}{a_w},$$  (A7)

$$p_{T,k+1}(z_i) = p_T(z_{i-1}) \left( \frac{p_d(z_i)}{p_d(z_{i-1})} \right)^{\beta_{T,k+1}(z_{i-1/2})},$$  (A8)

where $\beta_{T,k+1}(z_{i-1/2}) = \frac{T_d(z_i)+T_d(z_{i-1})}{T_b(z_i)+T_b(z_{i-1})} \cdot \frac{1+b_w \sqrt{V_{wT,k+1}(z_i)V_{wT}(z_{i-1})}}{1+2b_w \sqrt{V_{wT,k+1}(z_i)V_{wT}(z_{i-1})}}$.

At each altitude level $z_i$, the initial values for the iteration are ($k = 0$):

(1) $V_{wT,0}(z_i) = \frac{q_b(z_i)}{a_w+b_w q_b(z_i)}$ and $p_{T,0}(z_i) = p_d(z_i) - 0.2 \cdot c_{q2T} \cdot q_b(z_i) p_d(z_i) / T_d(z_i)$, if $T_d(z_i) \leq T_{dThres} \vee z_i = z_{iniMoist}$;

(2) $V_{wT,0}(z_i) = V_{wT}(z_{i-1})$ and $p_{T,0}(z_i) = p_T(z_{i-1}) \cdot (1 + |z_i - z_{i-1}|/H_0)$, if $T_d(z_i) > T_{dThres} \wedge z_i < z_{iniMoist}$, where $H_0 = 8$ km and $T_{dthres} = 240$ K.

The iteration ends at $k = k + 1$ that satisfies $\left|\left(V_{wT,k+1}(z_i) - V_{wT,k}(z_i)\right)/V_{wT,k}(z_i)\right| < (dV_w/V_w)_{tol}$, where $(dV_w/V_w)_{tol} = 0.01\%$ is the convergence tolerance, yielding $V_{wT}(z_i)$ and $p_T(z_i)$ as converged values. At all higher levels, $z_i > z_{iniMoist}$, we use the same formulations to assign $V_{wT}$ and $p_T$ as used at $z_{iniMoist}$, i.e., the initial-value formulations under iteration condition (1) above.

The reason that set a low-bounded value in Equation (A7), with $q_{minE} = 0.001$ g/kg, is because we try to prevent unphysical (negative) values in case $T_b < T_d$ occurs, which can happen within errors of $T_b$ and $T_d$ at upper troposphere levels where $q$ is very small (less than about 0.1 g/kg). We note that the error estimation is unaffected by this low-bounding as it does not depend on $q$ itself. Also, the resulting humidity profile after the optimal estimation step is receiving essentially negligible weight at the high tropospheric altitudes from this step 1b profile compared to the background humidity profile.

The reason to set a low-bounded value based on profile $V_{wT}(z)$ the retrieved specific humidity profile $q_T(z)$ can be computed using the inverse version of Equation (3) in the main text in the form:

$$q_T(z) = \frac{a_w V_{wT}(z)}{(1 + b_w V_{wT}(z))}.$$  (A9)

Using this equation together with Equation (7) in the main text, we can derive a linearized version (differential form) of $q_T$ related to $T_b$ and $T_d$. Using for the purpose the approximate assumptions $V_{wT} \approx q_T/a_w$ and $dp_d/p_d \approx dp_T/p_T$, which are reasonably valid over the moist air retrieval altitude range of interest, the linearized version reads:

$$dq_T = \left( \frac{2\frac{p_d}{p_T}T_b - T_d}{T_d} c_{T2q} \right) dT_b - \left( \frac{\frac{p_d}{p_T}T_b^2}{T_d^2} c_{T2q} \right) dT_d.$$  (A10)

The variance of the retrieved specific humidity $u_{qT}^2(z)$ can hence be calculated using the variance-based uncertainty propagation law as:

$$u_{qT}^2 = \left( \frac{2\frac{p_d}{p_T}T(z)_b - T_d(z)}{T_d(z)} c_{T2q} \right)^2 u_{Tb}^2(z) + \left( \frac{\frac{p_d(z)}{p_T(z)}T_b^2(z)}{T_d^2(z)} c_{T2q} \right)^2 u_{Td}^2(z).$$  (A11)

Similarly, based on Equations (8) and (15), we obtain for the linearized expression of $p_T(z)$:

$$dp_T(z) = \beta_T'(z) \frac{p_T(z)}{p_d(z)} dp_d(z),$$  (A12)

where $\beta_T(z) = \frac{T_d(z)(1+b_w V_{wT}(z))}{T_b(z)(1+2b_w V_{wT}(z))}$. Based on this single-term equation the uncertainty of $p_T(z)$ can be propagated in a straightforward manner from the dry pressure uncertainty $u_{pd}(z)$ via:

$$u_{pT}(z) = \beta_T(z)\left(\frac{p_T(z)}{p_d(z)}\right)u_{pd}(z). \tag{A13}$$

**Appendix B Bias Correction of Background Profiles and Its Effects**

Taking the advantage of the variables calculated in daily error fields, the bias-corrected background temperature profiles $T_b$ can be calculated as:

$$T_b = T_f - \Delta\overline{T}_{f-a}, \tag{A14}$$

where $T_f$ is co-located ECMWF forecast temperature, $\Delta\overline{T}_{f-a}$ is the bias-correction term obtained from bi-linear interpolation of $\Delta\overline{T}_{f-a}$ from the four surrounding grid points, where $\Delta\overline{T}_{f-a}$ at each grid point is calculated as the difference profile between mean forecast temperature and mean analysis temperature, $\Delta\overline{T}_{f-a} = \overline{T}_f - \overline{T}_a$. Similarly, the bias-corrected specific humidity profile is calculated as:

$$q_b = q_f - \Delta\overline{q}_{f-a}. \tag{A15}$$

Again, $\Delta\overline{q}_{f-a}$ is obtained from bi-linear spatial interpolation and $\Delta\overline{q}_{f-a}$ at each grid point is calculated as $\Delta\overline{q}_{f-a} = \overline{q}_f - \overline{q}_a$. Illustrations of the effects of bias-correction of background profiles are shown in Figures A1 and A2 below.

*Bias correction effects illustrated for individual-event temperature and humidity profiles*

In order to investigate the effects of bias-correction of background profiles on retrieval results, we compare the moist profiles retrieved using the bias-corrected background profiles and the profiles retrieved using the original background profiles. As a first example, we used three exemplary events from simMetOp, CHAMP, and COSMIC. The results are shown in Figure A1. From this result, and also from extensive further testing results, we find that bias-correction of background profiles is useful for enabling reduced biases also in retrieved profiles.

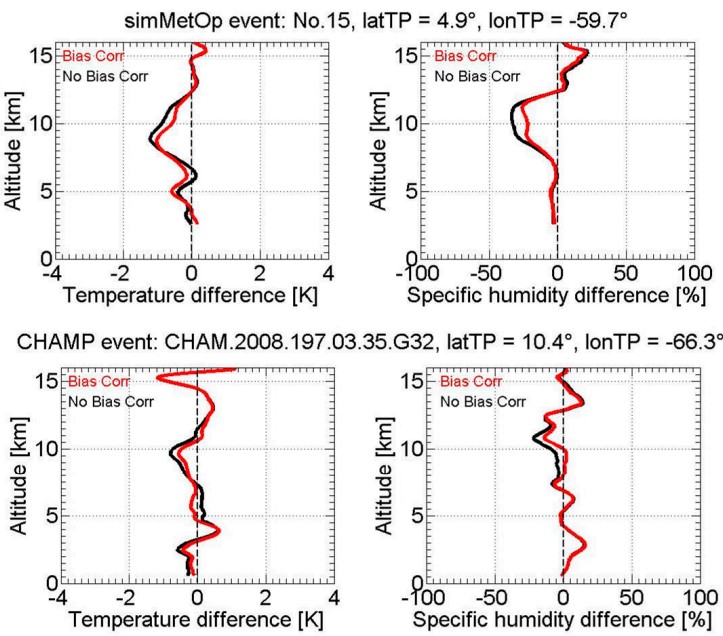

**Figure A1.** *Cont.*

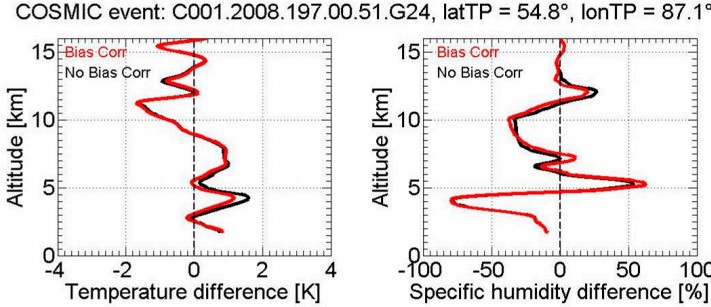

**Figure A1.** Differences between RO retrieved profiles and ECMWF co-located analysis profiles obtained from using the bias-corrected background profiles (red, "Bias Corr") and from using the original background profiles (black, "No Bias Corr") profiles for three exemplary RO events from simMetOp (upper), CHAMP (middle), and COSMIC (bottom) from 15 July 2008.

*Bias correction effects illustrated for test-day ensemble of temperature and humidity profiles*

The effects of the background bias correction scheme are investigated as well statistically, again by comparing the retrieval results between those obtained using the bias-correction retrieval results and those obtained without using the bias-correction. The uncertainties used for the retrieval examples illustrated here are the dynamic uncertainties. In Figure A2, from left to right panels, statistical results are shown for the simMetOp, CHAMP, and COSMIC missions.

In particular in tropical regions a good quality of humidity retrievals can be rather challenging so that bias correction is expected to be most helpful in such conditions. Indeed, from the results in Figure A2 we can see that the bias-correction scheme can obviously reduce the biases in retrieved moist profiles, especially for the humidity profiles in tropical regions, where the amount of moisture is significant and the humidity profiles are more readily biased.

**Figure A2.** *Cont.*

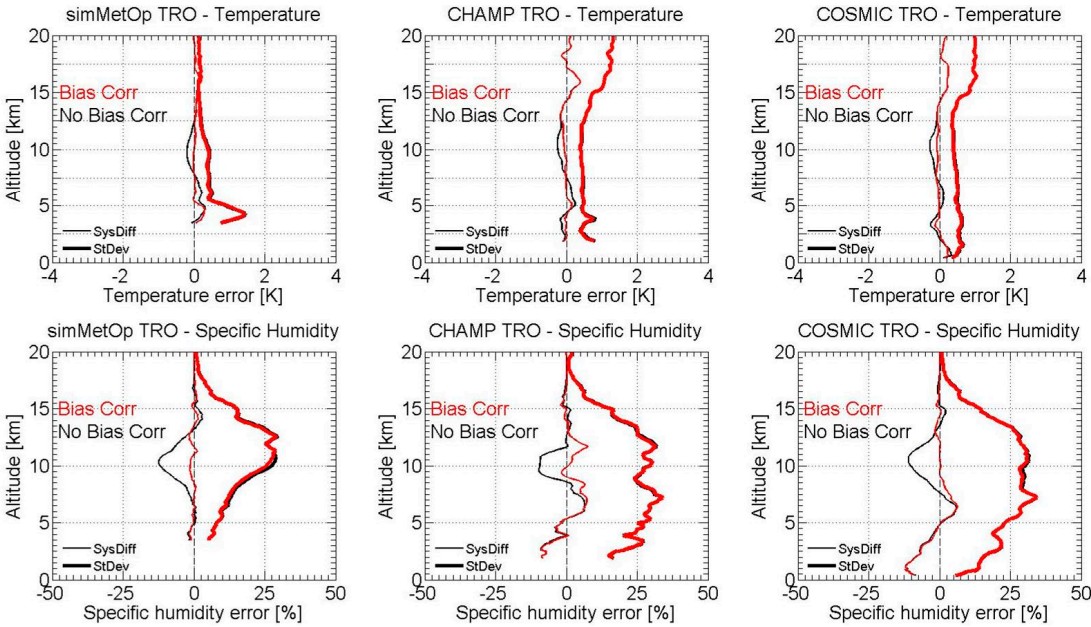

**Figure A2.** Systematic differences and standard deviations of moist temperature and specific humidity for simulated MetOp (left), CHAMP (middle), and COSMIC (right) events in the global domain (upper two rows) and TRO regions (bottom two rows). Statistics are shown for the bias-corrected case (Bias Corr) and the no bias corrected case (No Bias Corr).

## Appendix C  Vertical Correlations of Observations and Background Errors

The vertical correlations of the input parameters were also investigated in order to understand the level of approximation if they are disregarded in the current OPSv5.6 and dynamic approach implementations of the new algorithm. The error correlation matrix was calculated by constructing a global-mean error covariance matrix using all the ensemble of difference profiles between the forecast/observed and analysis profiles. By dividing the values in the error covariance matrix by the corresponding squared-uncertainty values (variances) in the matrix diagonal, the error correlation matrix can be obtained (e.g., [52]).

Figure A3 shows the correlation matrix (left), exemplary correlation function (middle), and correlation length (right) for the four input parameters, i.e., observed dry temperature, observed dry pressure, background temperature, and background specific humidity. The data shown are mainly from 15 July 2008. However, in order to show the variations of correlations with day of month, we also show the correlation functions and correlation lengths from the 5th and 25th of July.

Figure A3 shows that both correlation functions and correlation lengths show little variations with day of month. Correlation functions of all the four parameters are close to Gaussian shape at the main peak. From the main peak outwards, the correlation functions of observed dry temperature, background temperature, and observed dry pressure have some negative side peaks, while the functions of background specific humidity are all positive. Except the correlation lengths of observed pressure being slightly larger, with values varying around 2 km, the correlation lengths of the other three parameters are limited to about 1 km to 1.5 km.

These results indicate that except the observed pressure, the correlations of the other three parameters are not significant. Therefore, in the new algorithm as presented here, it was considered reasonable to disregard the correlations of the input variables within the scope of this study. Further advancements that include the full covariance formulation and propagation in the algorithm are described in a separate follow on paper, based on initial descriptions in Kirchengast et al. [40] and Schwarz [44].

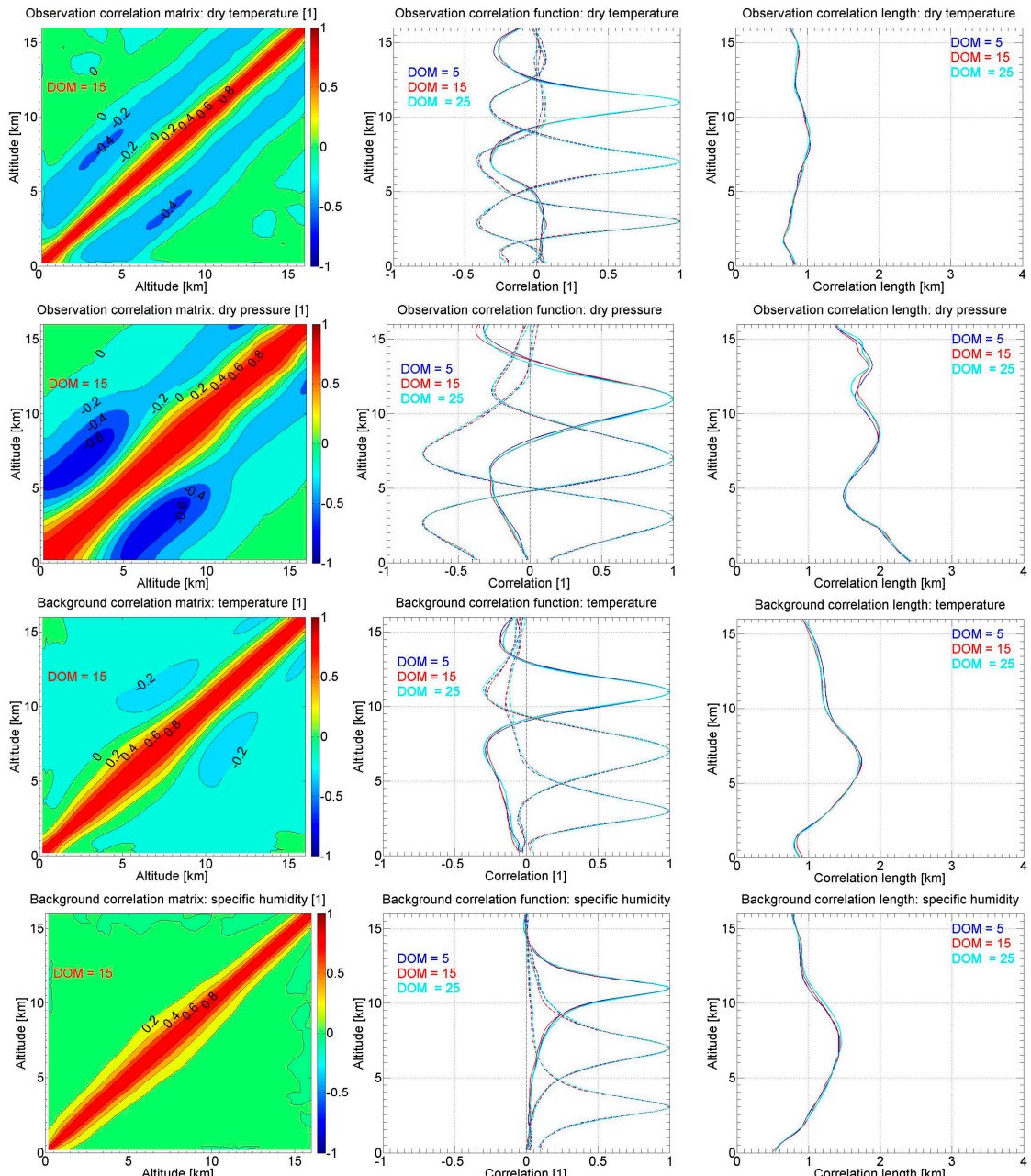

**Figure A3.** Correlation matrices (left), exemplary correlation functions at three exemplary altitude levels of 11 km, 7 km, and 3 km (middle), and estimated correlation length for correlation functions (right) for the observed dry temperature uncertainty (first row), observed dry pressure uncertainty (second row), background temperature uncertainty (third row), and background specific humidity uncertainty (fourth row). The correlation matrices are shown for 15th July 2008 only, and the correlation functions and correlation lengths are shown for 5th, 15th, and 25th of July 2008.

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
