# Peer review of "A New Algorithm for the Retrieval of Atmospheric Profiles from GNSS Radio Occultation Data in Moist Air and Comparison to 1DVar Retrievals"

_remotesensing, doi:10.3390/rs11232729_

Round 1

Reviewer 1 Report

The background, a description of the proposed algorithm, and interpretation of the results are fully presented. However, CHAMP and COSMIC data are considered to be too long ago, and further explanation of EGOPS is required.

Reviewer 2 Report

A new algorithm for the retrieval of atmospheric profiles from GNSS radio occultation data in moist air and comparison to 1DVar retrievals

Li et al. 

Summary

The authors describe an approach for deriving q, T, and P profiles from RO data which includes transparent derivation of these variables error estimates for the global atmosphere.  Such profiles are fundamental to atmospheric studies, climate monitoring and other applications and thus the work has much value if the method can improve upon or maintain data quality while simplifying the derivation of these variables.  Overall the method seems to provide similar quality q, T and P profiles as other 1DVar approaches but with greater ability for error estimation and bias correction. The paper is generally well written, however some aspects of the methodology need further clarification.  In addition, some aspects of the comparisons with other methods are too vague and do not discuss important differences between their results and those of other 1DVAR methods that have significant implications for users of such data sets.  Finally, it is not clear why the “dynamic” approach is being presented here alongside the “static” approach for error characterization of this new method since only the “static” approach is described in the methods section.  I recommend publication after the authors address these concerns. 

Major comments:

Lines 142-152: It is somewhat confusing that the dynamic approach is not described in this paper but is evaluated here alongside the static approach.  If the dynamic approach is to be presented in a future publication, as stated here, will that publication not evaluate the dynamic approach too?  I recommend removing the aspects of the manuscript that evaluate the dynamic approach, if it will be described in a separate publication and is not to be described here.  Please provide some sort of justification in the text for including the dynamic approach in the evaluation but not the methods section, especially if there are plans to publish this method at a later date.

Lines 254-260, Eq 12 & lines 273-277 and Eq. 16 - It is not clear how you have observed pd(z) and Td(z) at all levels z.  You state on lines 165-167 that only the ratio of these values is known from the refractivity observed with RO at every level.  So more steps must be needed to get Td and pd at each level.  Please explain.

Figure 2: The labeling is confusing.  The dash could be interpreted as a minus sign.  Please change title of columns 2 and 3.  Also, it would be better to remove column 3 since the dynamic approach will be presented in a separate publication and has not been described here (or elsewhere in the literature).  

Figure 3: Suggest removing dynamic approach results to keep the paper focused on what is described in this work.  Shorten figure capture and move necessary text to main body.

Figure 4&5: Suggest removing dynamic approach results for future publication.  Still unclear if it is published elsewhere (see lines 

Lines 527-529: This statement is not accurate.  The dynamic and static approaches show larger differences from CDAAC and ROM-SAF from about 4-7 km or so.  CDAAC and ROM-SAF differences are then larger from about 7-10 km.  

Figure 9 and lines 642-644: Actually, the larger systematic differences in specific humidity at low latitudes in the lower troposphere with both the static and dynamic approaches would have significant consequences for tropical studies using these data as opposed to the CDAAC or ROM-SAF data.  Global averages (top row) seem to reflect these large errors in the tropical lower tropospheric humidity for these new data sets.  This result needs to be mentioned in the text and recognized as a current issue with the algorithm being described.  Hopefully the authors have some ideas as to how to fix these errors in future versions?

Minor issues:

Line 23: Remove “these”  and “atmospheric”.

Line 25: Replace “in transparent steps” with something like “that is easily reproduced”.  It is awkward as is.

Line 33: Remove “-day“.

Line 49: Change “in” Low Earth… to “on” Low Earth…

Line 63: Remove “air” from Most air retrieval…I’m not sure what air retrieval algorithms are actually.  Do the authors mean Moist air retrieval algorithms?  Please clarify.

Line 114: Remove “are” at the beginning of the line.

Line 137: Please define WEGC.  Move OPS definition from line 140-141 to here where OPS is first used.

Line 149: Is “test-day” a well-known term for this community?  It does not seem like -day is needed.  Please define what is meant by “test-day”.

Line 345: Change to “In the future, the propagated individual-profile based observation uncertainties (and error correlation matrices) will be used in the rOPS system [43].”
